# VERISIMPL: Robust Optimization Modeling from Natural Language using Simplification-based Verification

**Sumaya Abdul Rahman** [1 2]   **Seckhen Ariel Andrade Cuellar** [3]   **Ghani Raissov** [3]   **Mohammad Raza** [4]

## Abstract

Natural language interfaces can greatly benefit the accessibility and usability of optimization modeling, and recent advances in large language models (LLMs) show promise in automatically translating textual problem descriptions into executable solver formulations. However, a key challenge for existing approaches is to ensure that the inferred formulation correctly implements the intended task, even if it may execute without errors. We introduce VERISIMPL, a solver–LLM framework for robust natural-language-to-optimization formalization. Our approach is based on the idea of *simplification-based verification*, where the optimization solver is leveraged to generate simplified diagnostic queries about a candidate formulation to allow the LLM to tractably reason about the correctness of the formulation with respect to the task description. We present such simplification strategies along different dimensions with respect to problem constraints and decision variables, which allow the LLM to reason locally under fixed global contexts. Evaluations on a range of optimization benchmarks show how our approach provides consistent improvements in accuracy over existing methods, while also providing a novel high-precision self-verification signal.

## 1. Introduction

Optimization plays a central role in decision-making across a wide range of industries, including manufacturing, logistics, supply chains, energy systems, healthcare, and finance (Sadana et al., 2025). Many of these real-world decision problems are naturally formulated as mathematical optimization models, such as Mixed-Integer Linear Programming (MILP) problems, which are one of the most widely used and expressive paradigms (Williams, 2009).

Despite their practical importance, optimization modeling remains difficult to access for non-expert users. Formulating a correct optimization problem typically requires deep expertise in operations research to translate a real-world scenario into mathematical constraints and objectives, as well as programming expertise to implement the formulation using solver-specific APIs such as Gurobi (Gurobi, 2026), CPLEX (IBM, 2026), or SCIP (Achterberg, 2009). This dual expertise requirement creates a significant barrier for many potential users, including small businesses, planners, and domain specialists, who may understand their operational needs well but lack the technical background required to formalize them.

Recent advances in Large Language Models (LLMs) offer a promising pathway toward lowering this barrier. By generating solver code from natural language descriptions of optimization problems, LLM-based interfaces have the potential to expand access to optimization technology and also improve usability for expert users (Liu et al., 2024). Initial approaches were based on direct prompting (Yang et al., 2024), while more advanced techniques such as agentic systems (Ahmaditeshnizi et al., 2024; Xiao et al., 2024) and fine-tuned domain-specific models (Tang et al., 2024; Jiang et al., 2025) have shown notable improvements in accuracy.

However, a key challenge remains: even when an LLM-based system produces solver code that may execute without errors, the generated formulation may not correctly formulate the intended problem, leaving the burden of verification entirely on the user in all cases. Optimization models are particularly prone to subtle semantic errors: a constraint may be slightly misspecified, an index mishandled, or an objective incorrectly aggregated. For example, Figure 1 illustrates an example optimization problem and two solver code formulations: one correct and one incorrect. While the two formulations appear very similar, the incorrect version contains subtle semantic mistakes in the objective function and one of the constraints that fundamentally change the meaning of the model. Such errors are difficult to detect and

Code available at: https://github.com/suabar/VeriSimple.git [1]Hamad Bin Khalifa University, Doha, Qatar [2]Texas A&M University at Qatar, Doha, Qatar [3]Carnegie Mellon University in Qatar, Doha, Qatar [4]Qatar Computing Research Institute, Doha, Qatar. Correspondence to: Mohammad Raza <mraza@hbku.edu.qa>.

*Proceedings of the 43rd International Conference on Machine Learning*, Seoul, South Korea. PMLR 306, 2026. Copyright 2026 by the author(s).

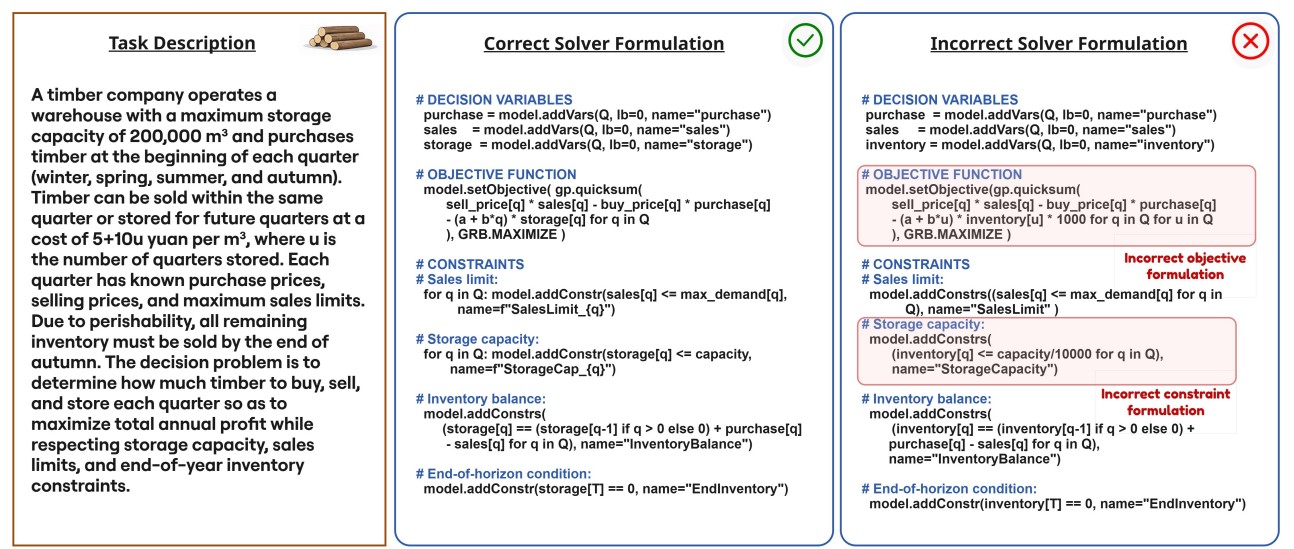

*Figure 1.* A sample optimization modelling problem. For a given natural language problem description, the LLM may generate the correct solver formulation or an incorrect one that looks very similar but contains subtle semantic errors that are difficult to detect.

can lead to invalid optimization results. While standard code generation approaches often utilise unit tests or behavioral test cases generated from the task specification, such approaches are infeasible for optimization code formulations: since correctness depends on global interactions between decision variables, constraints, and objectives, generating reliable test instances that satisfy feasibility or optimality conditions in high-dimensional spaces is itself a hard optimization problem.

In this work, we address this challenge by introducing a novel solver-LLM framework for natural language optimization modeling. Our approach not only improves the accuracy of NL-to-solver code translation but also provides a self-verification signal that indicates when there is high confidence in a generated formulation. Our method is based on a new paradigm of *simplification-based verification*. The key idea is that instead of relying on the LLM to generate verification tests, we leverage the optimization solver to construct simplified diagnostic queries about the original problem. These simplified queries reduce complexity and probe specific properties while preserving the global semantic structure of the original formulation. In particular, we reduce complexity along different problem dimensions represented by constraints and decision variables, allowing us to probe feasibility and optimality properties. Thus conceptually, our approach inverts the conventional verification workflow: instead of using the LLM to propose test scenarios that are then checked by the solver, we leverage the solver's reliability in constructing feasible and optimal solutions in high-dimensional space to obtain reduced-complexity problem instances that are tractable for reasoning by the LLM.

Our evaluation across four benchmark datasets spanning a range of optimization domains shows consistent improvements in end-to-end formalization accuracy over existing state of the art baseline methods. Moreover, our novel self-verification mechanism identifies a substantial subset of cases where the system can signal high confidence on the generated formulation, which can help to reduce the burden of manual inspection in many cases in practice.

In summary, we make the following key contributions in this work: (1) We introduce *simplification-based verification*, a framework for verifying natural language optimization models via solver-guided diagnostic problem simplifications that probe semantic correctness. (2) We present concrete simplification strategies along different dimensions of complexity represented by problem constraints and decision-variable dimensions, enabling tractable LLM-based reasoning about feasibility and optimality. (3) We provide an extensive empirical evaluation over four optimization benchmarks, demonstrating consistent improvements in accuracy over existing NL-to-solver systems, as well as a novel high-confidence self-verification signal.

## 2. Simplification-based Verification

In this section, we present a high-level overview of our simplification-based verification approach and we illustrate it with motivating examples. Given a natural language description of an optimization problem together with associated input data, the goal is to compute the optimal solution. Standard approaches generally follow a sequential process where the LLM or LLM-based agentic system first formulates the problem as code which is then executed by the

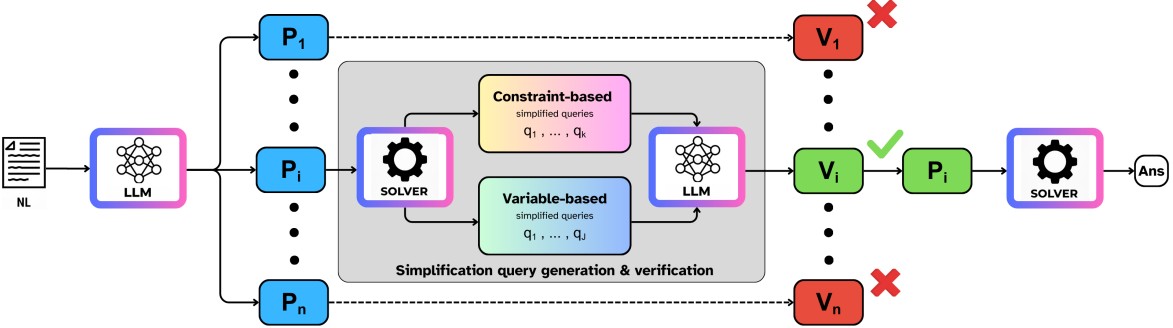

*Figure 2.* The simplification-based verification pipeline: using solver-generated simplified queries for effective LLM reasoning

optimization solver to compute the optimal solution. In contrast, our approach integrates the solver and the LLM in a deeper way, using them jointly for formulation verification rather than relying on the solver solely for execution.

**Outline.** Figure 2 illustrates the overall simplification-based verification pipeline. Our system first generates multiple candidate solver programs ($P_i$) using the LLM, where each $P_i$ denotes a candidate optimization program, such as the correct and incorrect programs shown for the timber example in Figure 1. Each program is then sent through a verification process, which starts with using the solver to generate a set of *simplification queries* ($q_i$) from the candidate program, where each $q_i$ denotes a solver-derived simplification query. For each simplification query, the solver computes a ground-truth result, and the LLM is asked to independently reason about the query and its expected outcome based only on the natural language problem description.

Hence simplification queries are not test cases in the standard sense of input-outputs that specify the expected behaviour of the program, but are instead properties that are derived from the program using the solver in order for the LLM to reason about the expected problem behaviour with respect to the task description. Consistency between the LLM's reasoning result and the solver's output yields a positive verification signal. Verification results ($V_i$) across multiple simplified queries are aggregated into final a verification score for each candidate program, which reflects how consistently the program aligns with the natural language specification. The program with the highest verification score is selected and executed on the input data to return the optimal solution.

**Simplification process.** Our method uses the solver to generate simplified problem instances along two key problem dimensions: constraints and decision variables. *Constraints-based simplification* focuses on validating whether individual constraints in the candidate program capture the semantics described in the natural language description. The key idea is to consider one constraint at a time while keeping others fixed, mutating it with respect to distinct semantic possibilities (strict satisfaction, boundary or violation) and generating concrete valuations for each possibility using the solver. These concrete valuations are then presented to the LLM, along with the natural language description of the problem, and the LLM is asked to reason about the feasibility of the valuation.

For example, for the storage capacity constraint in the incorrect program in Figure 1, the formulation wrongly enforces division of capacity (200000) by 10000, effectively requiring that inventory should always be less than 20 (which does not follow from the problem description). The solver will generate concrete valuations for the inventory according to this formulation, e.g. an inventory of 19 and 20 should be feasible, but 21 should be infeasible. When these simple queries with concrete inventory values are sent to the LLM, it can easily reason that all three inventory values should be acceptable under the problem description, and therefore the infeasibility query would fail verification. Thus the simple queries with concrete values allow the llm reasoning to detect this discrepancy and penalize the incorrect program.

While constraints-based verification checks for feasibility violations, it does not assess the optimality formulation of the candidate program. We address this verification aspect with *variable-based simplification*. While the full program has many interacting decision variables that must be mutually optimized, the key idea behind variable-based simplification is to reduce complexity by providing concrete values for many of these variables and only predict the remaining ones. Given a candidate program, we first run the solver to obtain the optimal valuation for all the decision variables, including the objective. We then produce simplified queries by providing the values for different subsets of variables and asking the LLM to predict only the remaining ones.

For instance, for the timber example in Figure 1, the objective function must maximise the total profit, but the incorrect program misformulates the objective calculation using a double for loop which adds storage costs multiple times for the same inventory. While asking the LLM to reason directly about the full problem and all unknown decision

variables is highly complex, when we provide concrete values for all of the decision variables except the total profit, then that effectively reduces the complexity of the task to only focus on the objective calculation. When the LLM reasons concretely given the values for all other variables (purchases, sales, and inventory), it correctly applies the storage cost once per quarter. This leads to the same total profit value inferred in the case of the correct program, but a differing value in the case of the incorrect program. Such discrepancy between concrete numerical reasoning and symbolic formulation evaluation serves as a strong error signal. In this way the simplification based on variable masking significantly reduces the reasoning complexity for the LLM to be able to detect such discrepancies.

Together, constraint-based and variable-based simplifications provide complementary verification signals that probe both feasibility and optimality aspects of candidate solver programs. By systematically simplifying the problem and comparing solver outputs with LLM reasoning grounded in the natural language description, our approach effectively detects subtle formulation errors that standard end-to-end generation methods fail to catch.

## 3. The VERISIMPL Algorithm

In this section, we formally present VERISIMPL as an abstract, solver- and LLM-agnostic algorithm that implements simplification-based verification for robust natural language to optimization formalization.

**Problem Setup.** We assume we are given a natural language specification of an optimization problem $x$ and associated structured input data $d$, which may be tabular data and parameters formatted in a structured format such as JSON or CSV. A candidate solver program $P$ (e.g. in Gurobi/C-PLEX/SCIP APIs) executing on data $d$ instantiates an optimization model $M(P, d) = (\mathcal{V}, \mathcal{C}, \mathcal{O})$ where $\mathcal{V}$ is the set of decision variables, $\mathcal{C} = \{c_1, \ldots, c_m\}$ the set of constraints and $\mathcal{O}$ is the objective function. We assume each indexed variable element is included as an atomic variable in the set $\mathcal{V}$, and this set also contains a special variable obj which represents the objective value. We define a valuation $v$ as a mapping of variables to values $v : \mathcal{V} \to \mathbb{R} \cup \mathbb{Z}$.

Our approach leverages two black-box components: an optimization solver $\mathcal{S}$ and a large language model $\mathcal{L}$. Given a model $M$, the solver interface is defined by $\text{SOLVE}(\mathcal{S}; M) \to (\texttt{status}, v)$, which returns a status and a possible valuation $v$ over $\mathcal{V}$. The $\texttt{status} \in \{\texttt{OPT}, \texttt{FEAS}, \texttt{INFEAS}\}$, where OPT indicates that $v$ is an optimal valuation, FEAS indicates that $v$ is feasible but not necessarily optimal, while INFEAS indicates that no satisfying valuation could be found.

The language model interface is defined by two main

---

**Algorithm 1** VERISIMPL$(\mathcal{L}, \mathcal{S}, x, d)$

1: $\{P_k\}_{k=1}^K \leftarrow \text{LLMGENPROGRAMS}(\mathcal{L}; x, d, K)$
2: **for** $k = 1 \ldots K$ **do**
3: $\quad M_k \leftarrow M(P_k, d)$
4: $\quad s_c \leftarrow \text{CONSTRAINTVERIFY}(\mathcal{L}, \mathcal{S}; x, d, M_k)$
5: $\quad s_d \leftarrow \text{VARVERIFY}(\mathcal{L}, \mathcal{S}; x, d, M_k, \{\{u\} \mid u \in \mathcal{V}\})$
6: $\quad s_f \leftarrow \text{VARVERIFY}(\mathcal{L}, \mathcal{S}; x, d, M_k, \{\mathcal{V}\})$
7: $\quad s_t, P_k \leftarrow \text{TYPEVERIFY}(\mathcal{L}; x, d, P_k)$
8: $\quad s(P_k) \leftarrow (s_c + s_d + s_f, s_t)$
9: **end for**
10: $k^\star \leftarrow \arg\max_k s(P_k)$
11: $(\texttt{st}, v^\star) \leftarrow \text{SOLVE}(\mathcal{S}; M(P_{k^\star}, d))$
12: **return** $(P_{k^\star}, \texttt{st}, v^\star)$

---

functions. Firstly, solver code generation is done by LLMGENPROGRAMS$(\mathcal{L}; x, d, K)$, which generates $K$ sample programs (using temperature sampling) for a given natural language task description $x$ and dataset $d$. The function uses standard error-based refinement to generate executable solver code (Shinn et al., 2023) (prompts shown in Figures 4 and 5 in Appendix). Secondly, the function LLMINFER$(\mathcal{L}, q) \to a$ uses the LLM to reason about a given verification query $q$ and infer its answer $a$, where verification queries are defined as follows.

**Verification queries.** A key abstraction in our approach is the use of *simplified verification queries*. Each query is a reduced problem derived from the original instance $(x, d)$ and a candidate program. We define two kinds of queries. Firstly, a **Feasibility query** $q^f = \text{FEASQUERY}(x, d, v)$ asks whether a given valuation $v$ satisfies the constraints described by $(x, d)$. The answer is a boolean $\hat{y} \in \{T, F\}$ that is true if the valuation is feasible. The LLM prompt for reasoning about feasibility queries is shown in Figure 6 in the Appendix. Secondly, a **Valuation query** $q^v = \text{VALQUERY}(x, d, v_{\mathcal{V} \setminus S}, S)$ asks the LLM to infer values for a subset of variables $S \subseteq \mathcal{V}$ given fixed values for the remaining variables. The answer is a (partial) valuation $\hat{v}_S$ over $S$. The LLM prompt for valuation queries is shown in Figure 7 in the Appendix.

**Algorithm Overview**. The main procedure is presented in Algorithm 1. Given the natural language problem description and data $(x, d)$, we first sample $K$ candidate programs from the language model (in our experiments we consider up to $K = 10$). For each candidate program, we perform three different kinds of verification which we describe in more detail below: constraint-based verification, variable-based verification and type verification. These produce four independent normalized scores corresponding to different properties of the candidate program: constraint verification ($s_c$), single-variable verification ($s_d$), full-variable verification ($s_f$), and type verification ($s_t$). The three semantic

**Algorithm 2** CONSTRAINTVERIFY$(\mathcal{L}, \mathcal{S}, x, d, M)$

1: pass $\leftarrow 0$; total $\leftarrow 0$
2: **let** $M = (\mathcal{V}, \mathcal{C}, \mathcal{O})$
3: **for all** $c \in \mathcal{C}$ **do**
4:  **for all** $\tau \in \{<, =, >\}$ **do**
5:   $(\text{st}, v) \leftarrow \text{SOLVE}(\mathcal{S}; \text{MUTATE}(M, c, \tau))$
6:   **if** st $\in \{\text{OPT}, \text{FEAS}\}$ **then**
7:    $q \leftarrow \text{FEASQUERY}(x, d, v)$
8:    $\hat{y} \leftarrow \text{LLMINFER}(\mathcal{L}, q)$
9:    pass $\leftarrow$ pass $+ \mathbb{1}[\hat{y} \Leftrightarrow (\tau \in \{<, =\})]$
10:    total $\leftarrow$ total $+ 1$
11:   **end if**
12:  **end for**
13: **end for**
14: **return** pass/total

---

**Algorithm 3** VARVERIFY$(\mathcal{L}, \mathcal{S}, x, d, M, T)$

1: pass $\leftarrow 0$
2: **let** $M = (\mathcal{V}, \mathcal{C}, \mathcal{O})$
3: $(\text{st}, v^\star) \leftarrow \text{SOLVE}(\mathcal{S}; M)$
4: **if** st $= \text{OPT}$ **then**
5:  **for all** $t \in T$ **do**
6:   $q \leftarrow \text{VALQUERY}(x, d, v^\star_{\mathcal{V} \setminus t}, t)$
7:   $\hat{v}_t \leftarrow \text{LLMINFER}(\mathcal{L}, q)$
8:   pass $\leftarrow$ pass $+ \mathbb{1}[\hat{v}_t = v^\star_t]$
9:  **end for**
10: **end if**
11: **return** pass$/|T|$

---

verification signals are aggregated using equal weighting to produce the final semantic verification score, while type verification is used only as a secondary tie-breaking signal through lexicographic ordering (thus we prioritize semantic correctness over syntactic consistency). The highest-ranked formulation is selected for final solving.

**Constraint-based Simplification** To assess whether a candidate program correctly captures the intended semantics of each constraint, we construct feasibility queries that probe the constraint under the distinct semantic possibilities that it presents (strict inequality, equality or violation) while fixing the remaining global context. We do this by mutating the constraint and generating witness valuations for these mutations using the solver. Each constraint $c \in \mathcal{C}$ is expressed in normalized form as $e(V) \leq 0$ for some expression $e(V)$ over variables $V \subseteq \mathcal{V}$. For a small fixed margin $\delta > 0$, we define three mutations:

$$c^< : e(V) \leq -\delta \qquad c^= : e(V) = 0 \qquad c^> : e(V) \geq +\delta$$

For $\tau \in \{<, =, >\}$ we define MUTATE$(M, c, \tau)$ to be the model $M$ where constraint $c$ is replaced by $c^\tau$. Algorithm 2 shows the full constraint verification process. For each con-

straint, we construct feasibility queries by using the solver to create satisfying valuations for the three mutation types. We then check each feasibility query with the LLM and the query passes if the LLM reasoning about feasibility of this valuation is consistent with the constraint implementation.

**Variable-based Simplification.** While constraint verification probes feasibility properties of individual constraints, it does not assess the constraint interactions and objective function logic that infers the optimal variable assignments. We approach this verification aspect with the idea of variable masking to construct simplified valuation queries. using the solver. Algorithm 3 describes this process. Given the task description and data, a candidate model $M$, and a collection $T$ of variable subsets to be masked, we first obtain an optimal valuation for $M$ using the solver. For each masking set $t \in T$, we then construct a valuation query that reveals the optimal values of all non-masked variables and asks the LLM to infer the values of the masked variables in $t$. The query is considered successful if the inferred values are consistent with the solver-derived optimal valuation.

In Algorithm 1, we apply this procedure to all singleton variable sets to query each variable individually (line 5), and to the full variable set $\mathcal{V}$ (line 6). The latter corresponds to reasoning about the complete optimization problem and thus provides a high-precision but low-coverage verification signal. For efficiency, our implementation also imposes a fixed upper bound on the number of singleton variables considered for masking.

**Type-based verification.** In addition to semantic verification, we also apply a lightweight type consistency check. The function TYPEVERIFY$(L; x, d, P)$ in Algorithm 1 prompts the LLM to reason about the domain of each decision variable declared in the program $P$ with respect to the problem description and data $(x, d)$. It returns a binary score $s_t$ that indicates if any type discrepancies are found, and also returns an updated program with the revised type annotations. The LLM prompt is shown in Figure 8 in the Appendix.

## 4. Evaluation

In this section we present an evaluation of our approach across multiple optimization modelling benchmarks and compare it against prior methods.

*Datasets.* We evaluate our method on four popular benchmark datasets for optimization modelling from natural language, spanning academic and industrial problems across different domains. **NL4Opt** consists of natural language descriptions of linear programming word problems and the corresponding formulations across multiple application domains, derived from the NL4Opt competition (Ramamonjison et al., 2023). We use the test split of the dataset which,

after removing some infeasible cases, contains 269 problems. **NLP4LP** is a dataset of 62 problems curated from optimization textbooks and lecture notes and includes both LP and MILP problems from domains such as facility location, network flow, scheduling, and portfolio management (Ahmaditeshnizi et al., 2024). **CompOR** (*Complex OR*) is a dataset of complex operations research problems sourced from academic literature and real-world industrial scenarios, characterized by longer descriptions and multi-constraint formulations (Xiao et al., 2024). We use the 17 problems that are currently publicly released and checked with feasible solutions. Finally, **IndOR** (*Industry OR*) is an industry-focused dataset consisting of 96 real-world problems across domains such as aviation, manufacturing, logistics, and energy (Tang et al., 2024).

*Baselines* We compare our approach with existing methods for optimization modeling from natural language using mainstream LLMs. Firstly, we evaluate direct LLM prompting to formulate optimization problems end-to-end without any additional reasoning frameworks. We use the strong general-purpose models GPT-4o (OpenAI, 2024) and R1 (Guo et al., 2025), where the models are provided only with the natural language problem description and data and required to generate solver code (prompt in Figure 4 in the Appendix). We also evaluate more advanced frameworks that incorporate base LLMs modularly. **OPTIMUS** is a structured agent-based framework for optimization modeling with general-purpose LLMs (Ahmaditeshnizi et al., 2024). It uses a sequential pipeline of multiple stages and tasks, calling specialized prompt-based agents for sub-tasks (e.g. formulator, coder, evaluator, manager). In our experiments, we ran the authors' released implementation of OPTIMUS with the LLM models that we use. We also compare to Chain-of-Experts (**CoE**) (Xiao et al., 2024), which is another multi-agent reasoning framework designed for solving complex operations research problems by decomposing them into expert-level subtasks. Another general-purpose code generation method that we consider is **SELFDEBUG**, which is an iterative self-refinement framework that enables LLMs to improve performance through execution feedback from past failures (Chen et al., 2024) which in our case are error messages from the solver (prompt used for error-based refinement in Figure 5 in Appendix).

## 4.1. Main results

Tables 1 and 2 report end-to-end accuracy across the four datasets, using GPT-4o and R1 respectively as the base model for all systems (except CoE, for which we report results directly from (Xiao et al., 2024)). Across both base models, VERISIMPL achieves the highest average accuracy, outperforming all prior methods. Furthermore, these gains are consistent across datasets: while some baselines exhibit strong performance on specific benchmarks but degrade

*Table 1.* Accuracy of systems, using GPT-4o as base (except CoE).

| System | NL4Opt | NLP4LP | CompOR | IndOR | Avg. |
|---|---|---|---|---|---|
| BASELLM | 75.5 | 43.5 | 64.7 | 42.7 | 56.6 |
| CoE | 58.9 | - | 25.9 | - | - |
| SELFDEBUG | 76.5 | 46.8 | 70.6 | 42.7 | 59.1 |
| OPTIMUS | 85.9 | 37.1 | 52.9 | 17.7 | 48.4 |
| VERISIMPL | **88.1** | **51.6** | **76.5** | **45.8** | **65.5** |

*Table 2.* Accuracy of systems, using R1 as base (except CoE).

| System | NL4Opt | NLP4LP | CompOR | IndOR | Avg. |
|---|---|---|---|---|---|
| BASELLM | 78.4 | 51.6 | 64.7 | 55.2 | 62.5 |
| CoE | 58.9 | - | 25.9 | - | - |
| SELFDEBUG | 82.9 | 56.5 | 70.6 | 61.5 | 67.9 |
| OPTIMUS | 84.4 | 32.2 | 47.1 | 56.3 | 55.0 |
| VERISIMPL | **88.8** | **58.1** | **76.5** | **67.7** | **72.8** |

substantially on others (e.g., OPTIMUS on NL4Opt versus IndOR), VERISIMPL consistently improves over all systems.

Comparing base models, average performance is higher with R1 than with GPT-4o, as expected given R1's stronger reasoning-oriented training. However, we note that with VERISIMPL we get almost the same performance for both models on NL4Opt (88.1 vs. 88.5), which closes the larger gap between the base models (75.5 vs. 78.4). Also, on CompOR both base models perform the same and see the same improvement with VERISIMPL, indicating that factors beyond explicit reasoning (e.g. differences in pretraining data or model architecture) may favor certain problem distributions.

Since benchmark dataset sizes vary substantially, Table 3 additionally reports dataset-size-weighted average accuracies. Similar conclusions can be seen for weighted averages: VERISIMPL remains the best-performing method across all benchmarks, with even larger gains over SELFDEBUG and BASELLM. We note that OPTIMUS sees a significant increase under weighted averaging due to its comparatively strong performance on the larger NL4Opt benchmark, although it still remains below VERISIMPL and SELFDEBUG overall.

Overall, the consistent improvements of VERISIMPL across distinct LLMs and benchmarks indicate the robustness of the simplification-based verification approach, as opposed to model-specific improvements from prompt engineering or fine-tuning. In Appendix C we also report evaluation using the Mistral large model (Mistral AI, 2024) which shows similar results.

## 4.2. Self-verification precision and coverage

Unlike prior systems, VERISIMPL is based on an approach of self-verification which can offer an additional indicator

*Table 3.* Comparison of unweighted and dataset-size-weighted average accuracies across benchmarks.

| System | Avg. (GPT-4o) | Weighted | Avg. (R1) | Weighted |
|---|---|---|---|---|
| BASELLM | 56.6 | 63.5 | 62.5 | 69.1 |
| SELFDEBUG | 59.1 | 64.8 | 67.9 | 74.1 |
| OPTIMUS | 48.4 | 63.1 | 55.0 | 69.6 |
| VERISIMPL | **65.5** | **73.4** | **72.8** | **79.5** |

*Table 4.* Self-verification precision and coverage of VERISIMPL

|  | VERISIMPL (GPT-4o) | | VERISIMPL (R1) | |
|---|---|---|---|---|
|  | Precision | Coverage | Precision | Coverage |
| NL4Opt | 92.7 | 32.1 | 95.7 | 38.2 |
| NLP4LP | 90.0 | 28.1 | 75.0 | 17.1 |
| CompOR | 100.0 | 53.8 | 66.7 | 15.3 |
| IndOR | 83.3 | 22.7 | 76.5 | 21.3 |
| **Avg** | 91.5 | 34.2 | 78.5 | 23.0 |

*Table 5.* Performance of ablation systems averaged over datasets

| System | GPT-4o | | | R1 | | |
|---|---|---|---|---|---|---|
|  | Acc. | Prec. | Cov. | Acc. | Prec. | Cov. |
| VERISIMPL | 65.5 | 91.5 | 34.2 | 72.5 | 78.5 | 22.8 |
| A-CONS | 63.9 | 75.7 | 78.9 | 71.4 | 77.8 | 76.0 |
| A-SINGLEVAR | 64.8 | 77.2 | 69.5 | 68.9 | 75.2 | 93.4 |
| A-FULLVAR | 62.2 | 74.0 | 27.7 | 69.9 | 77.5 | 66.7 |
| A-TYPE | 64.2 | 91.6 | 31.6 | 72.3 | 79.7 | 23.0 |

CONS performs only constraint-based verification (line 4 in Algorithm 1. **A-SINGLEVAR** performs only variable-based verification with singleton variable masks (line 5 in Algorithm 1), while **A-FULLVAR** performs only variable-based verification with all masked variables (line 6 in Algorithm 1. Finally, **A-TYPE** disables type-based refinement (line 7).

Table 5 shows the results of our ablation experiments for the both GPT-4o and R1. The results highlight the complementary roles of different verification signals consistently across both models. Overall, removing components reduces accuracy and verification precision, indicating that no single verification signal is sufficient to robustly assess semantic correctness. In particular, A-CONS shows that constraint feasibility alone is a weak proxy for correct problem interpretation, while A-SINGLEVAR and A-FULLVAR demonstrate that relying on LLM reasoning for optimal valuations alone provides insufficient robustness. A-TYPE highlights the benefits of type-level grounding and refinement in improving overall accuracy, although these gains are relatively smaller in comparison to the semantic verification components.

As for coverage, we see that A-SINGLEVAR and A-FULLVAR exhibit higher coverage rates at the cost of lower precision, which can be expected from the reduced overall constraints that they impose. By contrast, A-FULLVAR shows consistently low coverage, reflecting the brittleness of end-to-end objective verification: while effective at filtering incorrect formulations, such tasks are difficult for LLMs to reason about and can lead to rejection of many otherwise correct solutions.

Figure 3 shows the accuracy gains for the ablation systems on each of the benchmark datasets. We observe that there are varying patterns for the different benchmark domains, showing that no one verification feature performs consistently best across the different domains. However, A-FULLVAR mostly performs worse than the other features as can be expected from its lower coverage.

### 4.4. Qualitative analysis

In Appendices A and B, we provide detailed cases illustrating successfully verified formulations as well as verification failure cases. Success cases illustrate how solver-generated

of system confidence apart from the answer that the system returns, which may help reduce manual verification burden in practice. We investigated the performance of full self-verification by considering the cases where all verification queries in Algorithm 1 succeeded. Table 4 reports self-verification *precision* and *coverage* for VERISIMPL under GPT-4o and R1, where precision is defined as the proportion of correctly answered cases out of all fully verified cases, and coverage is defined as the proportion of fully verified cases out of all correctly answered cases.

Two trends stand out. First, precision is substantially higher for both models than the corresponding end-to-end accuracy shown in Tables 1 and 2, indicating that when all self-verification checks succeed, the resulting solutions are markedly more reliable. In particular, for GPT-4o, precision exceeds 90% on average, and on NL4Opt precision is above 90% for both models, highlighting that verification provides a strong confidence indicator in many cases. Second, coverage is comparatively low across benchmarks, which reflects the strict consistency requirements imposed by simplification-based verification: the model must align its natural language reasoning with all solver-grounded diagnostic queries, and any mismatches (whether due to problem description ambiguity or reasoning noise) can cause otherwise-correct solutions to fail full verification (false negatives). However, we note that even such limited coverage of 20–30% can provide a substantial reduction in manual verification burden in practice.

### 4.3. Ablation studies

We performed an ablation study to examine the effects of different aspects of our simplification-based verification approach. We define restricted variants of the VERISIMPL algorithm that isolate individual verification signals. **A-**

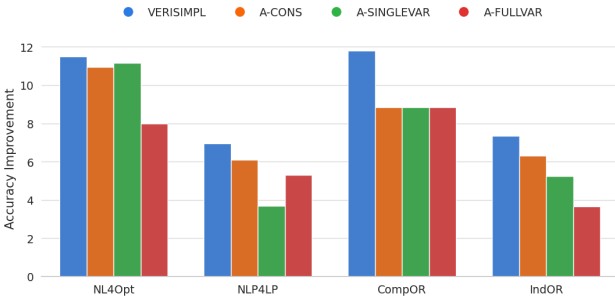

*Figure 3.* Accuracy improvement over the base model across datasets.

simplifications expose meaningful local and global structure, allowing the LLM to reason accurately about feasibility and optimality across different canonical optimization problem families, including coverage and facility selection, resource allocation, combinatorial planning with logical constraints, and routing and transportation problems. These cases show how consistency between solver witnesses and LLM reasoning provides a strong signal of formulation correctness.

The failure cases illustrate the rare instances when verification may pass on incorrect answers. The main reason for this is a *misinterpretation* of the natural-language specification that can lead both the symbolic formulation and the LLM's concrete reasoning to consistently agree on the same incorrect model of the problem, e.g. a shared incorrect decision-variable interpretation of "start time" in a shift scheduling problem, or not considering certain costs in a profit objective calculation. In these instances, the optimization model was internally coherent, and verification succeeded because the LLM's reasoning aligned with the solver's outputs due to the shared misinterpretation. Theses cases highlight limitations certain aspects not being explicitly handled by current verification: decision variables are assumed as given and shared between the llm reasoning and solver analysis, and simplification queries do not cover any aspects of the NL description that may be completely missed or ignored by the candidate program rather than incorrectly formulated. These limitations suggest directions for future work to expand verification to include variable definitions and coverage-oriented checks to detect unmodeled aspects of the natural-language description.

## 5. Related Work

**Optimization modeling from natural language.** Recent work has explored the use of LLMs to solve optimization problems from natural language descriptions. Some techniques are based on directly prompting the LLM to solve the full optimization problem (Yang et al., 2024), though scalability to more complex problems is a challenge for such approaches. To this end, many works have focussed on

translating natural language descriptions of problems into formal code formulations that can be executed by existing optimization solvers (Ramamonjison et al., 2023; Li et al., 2023). The key challenge here is to ensure that the formulation correctly implements the intended task, and recent works have explored more advanced techniques. Agentic systems employ multiple interacting agents that play different roles such as decomposing the task into intermediate steps, or iteratively refine, critique, or cross-check generated formulations, leading to improved performance over single-pass prompting (Shinn et al., 2023; Ahmaditeshnizi et al., 2024; Xiao et al., 2024). On the other hand, fine-tuning LLMs trained on selected optimization datasets strengthens adherence to solver APIs and common modeling patterns (Tang et al., 2024; Jiang et al., 2025). Despite these advances, subtle semantic errors such as wrongly formulated constraints or objectives continue to persist and are difficult to detect, especially for non-expert users. To address this gap, our work adopts a verification-centric perspective, using the optimization solver itself to probe the correctness of candidate formulations by generating simplified diagnostic queries for tractable LLM reasoning. We show how this approach not only improves overall accuracy, but also provides a high-confidence self-verification signal.

**Verification in LLM-based code generation.** Recent work on improving the robustness of LLM code generation increasingly adopts a verification-centric view, refining candidate programs using execution-based feedback rather than relying on single-shot generation. Prominent approaches run generated code against unit tests—often LLM-generated—and use compiler or runtime errors to guide self-debugging and self-refinement (Chen et al., 2024; Shinn et al., 2023). However, weak or overfitted tests are now recognized as a key limitation, motivating stronger adequacy criteria such as mutation-based test strengthening (Ravi et al., 2025) as well as metamorphic and property-based testing, which check semantic invariants when explicit oracles are unavailable (Wang & Zhu, 2024; Bose, 2025). Similar approaches have also been explored for code generation in reasoning domains, such as strengthening formalizations with logical instantiations inferred by the LLM (Raza & Milic-Frayling, 2025; Kalyanpur et al., 2024).

Our work positions in this general area with the main distinction being in domain and verification strategy. We focus on the optimization modeling domain, where generating reliable tests from the natural language specification is itself a hard optimization problem. We address this by leveraging the solver itself to generate simplified diagnostic queries from the candidate program, including constraint mutations and variable-masking under solver-optimal contexts, to allow tractable reasoning by the LLM. This inverts the standard "LLM-proposes-tests" workflow and is conceptually closest to mutation testing (DeMillo et al., 1978; Jia & Har-

man, 2011), with the main distinction being that the solver not only supplies the oracle, but also constructs semantically targeted simplifications that enable high-precision self-verification in this domain. Similar to the optimization domain, code-based tool-augmented approaches are also prevalent in related areas such as planning (Kambhampati et al., 2024; Guan et al., 2024), logical reasoning (Ye et al., 2023; Pan et al., 2023) and auto-formalization (Wu et al., 2022; Jiang et al., 2023), where LLMs are used in conjunction with powerful backend solvers, planners or theorem provers. While our focus here has been on the optimization domain, the underlying principle of solver-guided verification query generation and simplification could in principle be applicable in other such domains, and this will be interesting exploration for future work.

# 6. Conclusion

We introduced simplification-based verification, a novel solver–LLM framework for natural-language-to-optimization modeling. By leveraging solver-generated simplified queries to enable tractable LLM reasoning, our method improves formulation accuracy and also provides a high-precision self-verification signal. A comprehensive evaluation across different benchmark datasets and base models demonstrates consistent gains over prior methods and highlights the effectiveness of our method for robust optimization modeling from natural language.

# Impact Statement

This paper presents work whose goal is to improve the reliability of natural language interfaces for optimization modeling. By reducing silent semantic errors in automatically generated optimization formulations, the proposed approach may help make advanced optimization tools more accessible to non-expert users without formal training in operations research or solver programming, while also supporting expert users in rapidly prototyping and validating optimization models. The methods are general-purpose and intended for decision-support settings, and they do not introduce new ethical concerns beyond those commonly associated with machine learning and AI-assisted optimization technologies. Accordingly, we do not identify broader societal risks that require specific discussion beyond standard considerations.

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

# A. Verified Success Cases

## A.1. Success Case 1: Chain Store Location Problem

**Problem Description.** A convenience supermarket is planning to open several chain stores in a newly built residential area in the northwest suburb of the city. For shopping convenience, the distance from any residential area to one of the chain stores should not exceed $800\,\mathrm{m}$. Table 6 shows the new residential areas and the residential areas within a radius of $800\,\mathrm{m}$ from each of them.

**Question.** What is the minimum number of chain stores the supermarket needs to build among the mentioned residential areas, and in which residential areas should they be built?

*Table 6.* Residential areas covered within an $800\,\mathrm{m}$ radius

| Area Code | Residential Areas within $800\,\mathrm{m}$ Radius |
|-----------|---------------------------------------------------|
| A | A, C, E, G, H, I |
| B | B, H, I |
| C | A, C, G, H, I |
| D | D, J |
| E | A, E, G |
| F | F, J, K |
| G | A, C, E, G |
| H | A, B, C, H, I |
| I | A, B, C, H, I |
| J | D, F, J, K, L |
| K | F, J, K, L |
| L | J, K, L |

**Gurobi Formulation and Verified Solution.** The problem is formulated as a binary set covering model, where a binary decision variable indicates whether a store is built in a residential area. The objective is to minimize the number of stores while ensuring that every residential area is covered by at least one store within $800\,\mathrm{m}$.

```
import json
import gurobipy as gp
from gurobipy import GRB

DATA_FILE = '../IndustryOR/10/data.json'
with open(DATA_FILE, 'r') as f:
    data = json.load(f)

area_codes = data['area_codes']
coverage = data['coverage']

model = gp.Model()

# Decision variables: y[i] = 1 if a store is built in area i
y = model.addVars(area_codes, vtype=GRB.BINARY, name='y')

# Objective: Minimize the number of stores
model.setObjective(gp.quicksum(y[i] for i in area_codes), GRB.MINIMIZE)

# Coverage constraints
for area in area_codes:
    model.addConstr(
```

```
            gp.quicksum(y[j] for j in coverage[area]) >= 1,
            f'Cover_{area}'
    )

model.optimize()
```

**Verified Optimal Solution.**  The optimizer returned the following solution:

- Stores built in areas: {A, I, J}

- Total number of stores: 3

**LLM Masked Reasoning and Verification Outcome.**  Below is the reasoning produced by our VERISIMPL model. The model reasons correctly over the coverage constraints and computes the correct optimal value, which is subsequently validated by the solver-based verifier.

> "The problem requires minimizing the number of chain stores while ensuring that every residential area is within 800 meters of at least one store. Based on the known values, stores are already planned in areas A, I, and J. From the coverage table, area A covers A, C, E, G, H, and I; area I covers A, B, C, H, and I; and area J covers D, F, J, K, and L. Together, these three stores provide coverage for all residential areas (A through L). Therefore, the minimum number of stores required is 3, which matches the number of stores already planned."

The verifier confirmed that the predicted value `TotalStores` = 3 matched the solver-optimal objective and that all coverage constraints were satisfied. Verification therefore passed.

### A.2. Success Case 2: Environmentally Friendly Transportation Planning

**Problem Description.**  A company plans to transport goods between the city and the suburb and needs to choose the most environmentally friendly transportation method. The company can choose from three transportation methods: motorcycle, small truck, and large truck. Each motorcycle trip produces 40 units of pollution, each small truck trip produces 70 units of pollution, and each large truck trip produces 100 units of pollution. The objective is to minimize total pollution.

The company can only choose two out of these three transportation methods. Due to road restrictions, the number of motorcycle trips cannot exceed 8. Each motorcycle trip can transport 10 units of products, each small truck trip can transport 20 units of products, and each large truck trip can transport 50 units of products. The company needs to transport at least 300 units of products. The total number of trips must not exceed 20.

**Gurobi Formulation and Verified Solution.**  The problem is formulated as a mixed-integer linear program. Integer variables represent the number of trips for each transportation method, while binary variables indicate whether a method is selected. The objective minimizes total pollution subject to transportation capacity, method selection, and trip limit constraints.

```
import json
import gurobipy as gp
from gurobipy import GRB

DATA_FILE = '../IndustryOR/39/data.json'
with open(DATA_FILE, 'r') as f:
    data = json.load(f)

transport_methods = data['transport_methods']
pollution = data['pollution']
max_methods = data['max_methods']
motorcycle_trip_limit = data['motorcycle_trip_limit']
transport_capacity = data['transport_capacity']
```

```
product_transport_requirement = data['product_transport_requirement']
total_trip_limit = data['total_trip_limit']

model = gp.Model()

trips = model.addVars(transport_methods, vtype=GRB.INTEGER, lb=0, name='trips')
use_method = model.addVars(transport_methods, vtype=GRB.BINARY, name='use_method')

model.setObjective(
    gp.quicksum(pollution[i] * trips[i] for i in range(transport_methods)),
    GRB.MINIMIZE
)

model.addConstr(gp.quicksum(use_method[i] for i in range(transport_methods)) == max_methods)
model.addConstr(trips[0] <= motorcycle_trip_limit)
model.addConstr(
    gp.quicksum(transport_capacity[i] * trips[i] for i in range(transport_methods))
    >= product_transport_requirement
)
model.addConstr(gp.quicksum(trips[i] for i in range(transport_methods)) <= total_trip_limit)

for i in range(transport_methods):
    model.addConstr(trips[i] <= total_trip_limit * use_method[i])

model.optimize()
```

**Verified Optimal Solution.**    The solver returned the following optimal solution:

- Number of trips: $(0, \ 2, \ 0)$

- Total pollution: $600$

**LLM Masked Reasoning and Verification Outcome.**    Below is the reasoning produced by our VERISIMPL model, which correctly infers the masked decision variable and is consistent with the solver-optimal solution.

"The company aims to minimize total pollution while meeting the constraints. Each motorcycle trip produces 2 units of pollution and can transport 2 units of products, while each small truck trip produces 3 units of pollution and can transport 3 units of products. The company needs to transport at least 5 units of products, and the total number of trips must not exceed 4. Additionally, the number of motorcycle trips cannot exceed 3. The known total pollution is 6.0 units. If no motorcycle trips are made (trips[0] = 0), the pollution would entirely come from small truck trips. Since each small truck trip produces 3 units of pollution, 6.0 units of pollution corresponds to 2 small truck trips. This satisfies the product transport requirement ($2 \times 3 = 6$ units, which is greater than the 5-unit requirement) and the total trip limit ($2 \leq 4$). Therefore, trips[1] = 2."

The verifier confirmed that the predicted value for the masked variable trips[1] $= 2$ matched the solver-optimal solution and that all constraints were satisfied. Verification therefore passed.

**A.3. Success Case 3: Family Trip Selection with Logical Constraints**

**Problem Description.**    The Zhang family has six children: Harry, Hermione, Ron, Fred, George, and Ginny. The cost of taking Harry is $1200, Hermione is $1650, Ron is $750, Fred is $800, George is $800, and Ginny is $1500. The objective is to minimize the total cost of taking children on an upcoming trip.

The family can take at most four children and must take at least three children. Ginny is the youngest child and must be taken. If the couple takes Harry, they will not take Fred or George. If they take George, they must also take Fred and Hermione.

**Gurobi Formulation and Verified Solution.** The problem is formulated as a binary integer program. Each child is associated with a binary decision variable indicating whether they are taken on the trip. The objective minimizes total cost subject to cardinality, exclusion, dependency, and mandatory-selection constraints.

```
import json
import gurobipy as gp
from gurobipy import GRB

DATA_FILE = '../IndustryOR/53/data.json'
with open(DATA_FILE, 'r') as f:
    data = json.load(f)

n_children = data['n_children']
children_names = data['children_names']
cost = data['cost']
max_children = data['max_children']
min_children = data['min_children']
mandatory_children = data['mandatory_children']
exclusions = data['exclusions']
dependencies = data['dependencies']

model = gp.Model()

x = model.addVars(n_children, vtype=GRB.BINARY, name='x')

model.setObjective(
    gp.quicksum(cost[i] * x[i] for i in range(n_children)),
    GRB.MINIMIZE
)

model.addConstr(gp.quicksum(x[i] for i in range(n_children)) >= min_children)
model.addConstr(gp.quicksum(x[i] for i in range(n_children)) <= max_children)

for child in mandatory_children:
    idx = children_names.index(child)
    model.addConstr(x[idx] == 1)

for exclusion in exclusions:
    if_idx = children_names.index(exclusion['if'])
    for not_child in exclusion['not']:
        not_idx = children_names.index(not_child)
        model.addConstr(x[if_idx] + x[not_idx] <= 1)

for dependency in dependencies:
    if_idx = children_names.index(dependency['if'])
    for must_child in dependency['must']:
        must_idx = children_names.index(must_child)
        model.addConstr(x[if_idx] <= x[must_idx])

model.optimize()
```

**Verified Optimal Solution.** The solver returned the following optimal solution:

- Selected children: {Ron, Fred, Ginny}

- Total cost: $3050

**LLM Masked Reasoning and Verification Outcome.** Below is the reasoning produced by our VERISIMPL model for a masked variable setting.

> "The Zhang family must take at least one child, and C is mandatory, so C is already selected. The cost of taking C is $4. Since the family can take up to two children, we need to evaluate the possible combinations of children that include C and satisfy the constraints. If they take A, they cannot take B due to the exclusion rule. If they take B, they must also take A due to the dependency rule, but this would result in taking three children, which exceeds the maximum of two allowed. Therefore, the only feasible option is to take C alone. The total cost is thus $4, which is the cost of taking C."

The verifier confirmed that the predicted value for the masked variable `TotalCost` matched the solver-optimal objective value and that all logical and cardinality constraints were satisfied. Verification therefore passed.

### A.4. Success Case 4: Animal Feed Cost Minimization Problem

**Problem Description.** Suppose a certain animal needs at least $700\,\mathrm{g}$ of protein, $30\,\mathrm{g}$ of minerals, and $100\,\mathrm{mg}$ of vitamins daily. There are five types of feed available, and the nutritional content and price per kilogram of each type of feed are shown in Table 7. The objective is to formulate a linear programming model that meets the animal's growth needs while minimizing the total feed cost.

*Table 7.* Nutritional content and price of available feeds

| Feed | Protein (g) | Minerals (g) | Vitamins (mg) | Price (¥/kg) |
|------|-------------|--------------|---------------|--------------|
| 1 | 3 | 1 | 0.5 | 0.2 |
| 2 | 2 | 0.5 | 1 | 0.7 |
| 3 | 1 | 0.2 | 0.2 | 0.4 |
| 4 | 6 | 2 | 2 | 0.3 |
| 5 | 18 | 0.5 | 0.8 | 0.8 |

**Gurobi Formulation and Verified Solution.** The problem is modeled as a linear program with continuous decision variables representing the quantity (in kilograms) of each feed type. The objective minimizes the total cost while satisfying minimum nutritional requirements.

```
import json
import gurobipy as gp
from gurobipy import GRB

DATA_FILE = '../IndustryOR/30/data.json'
with open(DATA_FILE, 'r') as f:
    data = json.load(f)

n_feeds = data['n_feeds']
protein_requirement = data['protein_requirement']
minerals_requirement = data['minerals_requirement']
vitamins_requirement = data['vitamins_requirement']
feeds = data['feeds']

model = gp.Model()

x = model.addVars(n_feeds, vtype=GRB.CONTINUOUS, lb=0, name='x')
```

```
model.setObjective(
    gp.quicksum(feeds[i]['price'] * x[i] for i in range(n_feeds)),
    GRB.MINIMIZE
)

model.addConstr(
    gp.quicksum(feeds[i]['protein'] * x[i] for i in range(n_feeds)) >= protein_requirement
)
model.addConstr(
    gp.quicksum(feeds[i]['minerals'] * x[i] for i in range(n_feeds)) >= minerals_requirement
)
model.addConstr(
    gp.quicksum(feeds[i]['vitamins'] * x[i] for i in range(n_feeds)) >= vitamins_requirement
)

model.optimize()
```

**Verified Optimal Solution.**    The solver returned the following optimal solution:

- Feed quantities (kg): $(0.0,\ 0.0,\ 0.0,\ 39.7436,\ 25.6410)$

- Total cost: ¥32.4359

**LLM Masked Reasoning and Verification Outcome.**    Below is the reasoning produced by our VERISIMPL model, which reasons correctly over the nutritional constraints and computes the correct optimal value.

"The animal's daily nutritional requirements are 700 g of protein, 30 g of minerals, and 100 mg of vitamins. The feeds used are Feed 3, Feed 4, and Feed 5, as indicated by the known quantities of 39.74358974358974 kg and 25.641025641025642 kg for Feed 4 and Feed 5, respectively. The total cost of the feeds is ¥32.43589743589744. Using the price per kg for Feed 4 (¥0.3/kg) and Feed 5 (¥0.8/kg), their costs are calculated as 39.74358974358974 × 0.3 = ¥11.923076923076922 and 25.641025641025642 × 0.8 = ¥20.512820512820514, respectively. The sum of these costs is ¥32.435897435897436, which matches the total cost. Therefore, Feed 3 is not used, and its quantity is 0.0 kg."

The verifier confirmed that the predicted value for the masked variable feed_quantities[2] = 0.0 matched the solver-optimal solution and that all nutritional constraints were satisfied. Verification therefore passed.

**A.5. Success Case 5: Traveling Salesman Problem**

**Problem Description.**    A traveling salesman must visit seven customers located at seven different locations. The symmetric distance matrix between the locations is shown in Table 8. The salesman must start and end the tour at location 1, visit each location exactly once, and minimize the total travel distance.

*Table 8.* Distance matrix for the traveling salesman problem

|   | 1 | 2 | 3 | 4 | 5 | 6 | 7 |
|---|---|---|---|---|---|---|---|
| 1 | – | 86 | 49 | 57 | 31 | 69 | 50 |
| 2 | 86 | – | 68 | 79 | 93 | 24 | 5 |
| 3 | 49 | 68 | – | 16 | 7 | 72 | 67 |
| 4 | 57 | 79 | 16 | – | 90 | 69 | 1 |
| 5 | 31 | 93 | 7 | 90 | – | 86 | 59 |
| 6 | 69 | 24 | 72 | 69 | 86 | – | 81 |
| 7 | 50 | 5 | 67 | 1 | 59 | 81 | – |

**Gurobi Formulation and Verified Solution.**    The problem is formulated as a mixed-integer linear program using binary decision variables to indicate travel between locations. Subtour elimination constraints are enforced using the Miller–Tucker–Zemlin (MTZ) formulation to ensure a single Hamiltonian cycle.

```
import json
import gurobipy as gp
from gurobipy import GRB

DATA_FILE = '../IndustryOR/60/data.json'
with open(DATA_FILE, 'r') as f:
    data = json.load(f)

n_locations = data['n_locations']
distance_matrix = data['distance_matrix']

model = gp.Model()

x = model.addVars(n_locations, n_locations, vtype=GRB.BINARY, name='x')
u = model.addVars(n_locations, vtype=GRB.CONTINUOUS, lb=0, name='u')

model.setObjective(
    gp.quicksum(
        distance_matrix[i][j] * x[i, j]
        for i in range(n_locations)
        for j in range(n_locations)
        if i != j
    ),
    GRB.MINIMIZE
)

model.addConstrs(
    gp.quicksum(x[i, j] for j in range(n_locations) if i != j) == 1
    for i in range(n_locations)
)

model.addConstrs(
    gp.quicksum(x[j, i] for j in range(n_locations) if i != j) == 1
    for i in range(n_locations)
)

model.addConstrs(
    u[i] - u[j] + n_locations * x[i, j] <= n_locations - 1
    for i in range(1, n_locations)
    for j in range(1, n_locations)
    if i != j
)

model.addConstrs(x[i, i] == 0 for i in range(n_locations))

model.addConstr(gp.quicksum(x[0, j] for j in range(1, n_locations)) == 1)
model.addConstr(gp.quicksum(x[j, 0] for j in range(1, n_locations)) == 1)

model.optimize()
```

**Verified Optimal Solution.**    The solver returned the following optimal tour:

- Tour: $(1, 6, 2, 7, 4, 3, 5, 1)$

- Total distance: $153$

**LLM Masked Reasoning and Verification Outcome.**    Below is the reasoning produced by our VERISIMPL model for a masked tour variable.

> "The traveling salesman problem aims to minimize the total travel distance while visiting all locations exactly once and returning to the starting point. The known tour starts at location 1, proceeds to location 6, then to location 2, and ends back at location 1 after visiting all other locations. The total distance of the tour is given as 153.0. Using the distance matrix, the distances for the known segments of the tour are: $1 \rightarrow 6$ (69), $6 \rightarrow 2$ (24), and $4 \rightarrow 3$ (16). To complete the tour, the missing location (Tour[3]) must minimize the remaining distance while ensuring the total distance sums to 153. The remaining distances are calculated as follows: $2 \rightarrow$ Tour[3], Tour[3] $\rightarrow 4$, and $5 \rightarrow 1$. Testing all possible locations for Tour[3], location 7 fits the constraints and minimizes the total distance."

The verifier confirmed that the predicted value for the masked variable `Tour[3] = 7` matched the solver-optimal tour and that all routing and subtour-elimination constraints were satisfied. Verification therefore passed.

## B. Failure Cases

### B.1. Problem 52

#### B.1.1. PROBLEM DESCRIPTION

This problem is a workforce scheduling problem commonly encountered in public transportation operations. A 24-hour bus service requires a varying number of drivers and crew members across six consecutive 4-hour time periods, as summarized in Table 9. Drivers begin work only at the start of a time period and work continuously for 8 hours, thereby covering two consecutive periods. The objective is to determine the minimum number of drivers and crew members required to satisfy staffing requirements throughout the day. This problem is traditionally formulated as a shift-start scheduling problem, where decision variables represent the number of drivers starting their shifts at each time period.

*Table 9.* Daily Driver and Crew Requirements by Time Period

| Shift | Time Period | Required | Shift | Time Period | Required |
|-------|-------------|----------|-------|-------------|----------|
| 1 | 06:00–10:00 | 60 | 4 | 18:00–22:00 | 50 |
| 2 | 10:00–14:00 | 70 | 5 | 22:00–02:00 | 20 |
| 3 | 14:00–18:00 | 60 | 6 | 02:00–06:00 | 30 |

#### B.1.2. OBSERVED OUTCOME

When solved using our approach, the optimization model converged to an optimal solution with a total of 150 drivers and crew members. All numerical constraints were satisfied, and the solver reported a zero optimality gap. Furthermore, predictive inference verification (System-10) successfully reconstructed masked variables with 100% accuracy when all other values were fixed. Despite these indicators of correctness, this solution constitutes a false positive.

#### B.1.3. WHY THIS IS A FALSE POSITIVE

The false positive arises from a semantic mismatch in the interpretation of the decision variables.

In the correct formulation, each decision variable $x_i$ should represent the number of drivers starting their 8-hour shift at time period $i$. This definition is critical because each driver covers two consecutive 4-hour periods, creating overlap across time intervals. Consequently, staffing constraints must explicitly model this temporal coupling.

However, in the evaluated formulation, the decision variables were implicitly treated as the number of drivers assigned to or present in each individual time period, rather than the number of drivers starting their shifts. This interpretation removes the

continuity requirement and allows each time period to be satisfied independently, violating the operational constraint that drivers work continuously for 8 hours.

As a result, the model permits drivers to effectively "appear" and "disappear" between shifts, producing a solution that is mathematically feasible but operationally infeasible.

### B.1.4. FAILURE EXPLANATION

VERISIMPL failed to detect this modeling error because its predictive inference relies on algebraic consistency among fixed variables rather than semantic reasoning. When a variable was masked, its value was inferred by maintaining total consistency, without considering temporal coverage or shift overlap constraints. As a result, VERISIMPL produced correct numerical reconstructions despite the underlying model being structurally incorrect.

### B.1.5. ROOT CAUSE

This failure illustrates that semantic errors in decision-variable definitions can bypass both optimization solvers and inference-based verification mechanisms. The solver optimized the given formulation, and VERISIMPL verified internal consistency, but neither validated whether the decision variables correctly represented real-world operational behavior.

## B.2. Problem 86

### B.2.1. PROBLEM DESCRIPTION

This problem is a production planning problem from an industrial food manufacturing setting. Healthy Pet Foods Company produces two dog food products: *Meaties* and *Yummies*. Each product requires specific amounts of grains and meat per pack and incurs a variable cost for mixing and packaging. The company can sell all units it produces.

Monthly production is constrained by limited availability of raw materials and a capacity restriction on Meaties due to a specialized machine. The objective is to determine the optimal production quantities of Meaties and Yummies that maximize total profit, defined as sales revenue minus all production-related costs. The relevant product, cost, and resource data are summarized in Table 10.

*Table 10.* Healthy Pet Foods Production Data

| Parameter | Meaties | Yummies |
|---|---|---|
| Selling price (per pack) | $2.80 | $2.00 |
| Grains required (lbs/pack) | 2.0 | 3.0 |
| Meat required (lbs/pack) | 3.0 | 1.5 |
| Variable processing cost (per pack) | $0.25 | $0.20 |
| Monthly grain availability | 400,000 lbs | |
| Monthly meat availability | 300,000 lbs | |
| Meaties production capacity | 90,000 packs/month | |

### B.2.2. OBSERVED OUTCOME

When solved using our approach, the optimization model converged to a feasible and solver-optimal solution. The solver reported production quantities of 50,000 packs of Meaties and 100,000 packs of Yummies, and all resource constraints were satisfied. The solver terminated with zero optimality gap.

VERISIMPL successfully reconstructed the decision variables when masked, indicating strong numerical consistency across the solution. However, the reported objective value labeled as `TotalProfit` was inconsistent with the interpretation of profit described in the problem. Despite solver optimality, this instance constitutes a false positive.

### B.2.3. WHY THIS IS A FALSE POSITIVE

The false positive arises from a semantic misinterpretation in the formulation of the objective function.

In the correct formulation, profit must be computed by subtracting *all* production-dependent costs from revenue. These costs

include not only variable processing costs, but also raw material costs for grains and meat, which are explicitly priced in the problem description. Omitting any of these costs results in an objective that no longer represents true profit.

However, in the evaluated implementation of VERISIMPL, the objective function subtracted only the variable processing costs and failed to subtract raw material costs. As a result, the model maximized a gross margin quantity rather than true net profit. The solver therefore optimized an incorrectly defined objective while still satisfying all constraints.

This error caused the reported `TotalProfit` value and caused a mismatch between the solver's output and the economically correct profit implied by the problem description.

### B.2.4. FAILURE EXPLANATION

The failure in this case arises from the interaction between an incorrectly specified optimization objective and the behavior of VERISIMPL under variable masking.

When individual decision variables (e.g., production quantities of Meaties or Yummies) were masked, VERISIMPL inferred their values by maintaining consistency with the solver-reported objective value. Because the remaining variables were fixed, VERISIMPL effectively solved a constrained completion problem and reproduced the same (incorrect) objective value as the solver. In these cases, predictive inference passed because the inferred values were consistent with the solver's internally flawed objective.

However, when the aggregate variable `TotalProfit` itself was masked, VERISIMPL recomputed profit directly from the problem description. In this sampling instance, the system correctly subtracted all production-dependent costs, including raw material costs, as implied by the problem statement. This produced the correct profit value, which no longer matched the solver's objective output. Consequently, verification failed, not because VERISIMPL reasoned incorrectly, but because its correct reasoning conflicted with the solver's incorrectly defined objective.

In a subsequent sampling instance, however, the reasoning path taken by VERISIMPL differed. In that case, the system reconstructed `TotalProfit` using incorrect interpretation of the problem description and failed to subtract raw material costs, mirroring the same semantic error present in the optimization model. Because this incorrect reasoning produced a profit value that matched the solver's output, verification passed, resulting in a false positive.

### B.2.5. ROOT CAUSE

The root cause of this failure is a semantic inconsistency between the optimization model and the verification assumptions, compounded by reasoning during inference.

The optimization model omitted raw material costs from the objective function, causing the solver to maximize an incorrect notion of profit.

## C. Evaluation on Mistral model

We performed an evaluation using the Mistral large model (Mistral AI, 2024) as the base model. Table 11 shows the accuracy results for all systems across all benchmarks, and Table 12 shows the precision and coverage of verification of VERISIMPL with the Mistral model.

| System | NL4Opt | NLP4LP | CompOR | IndOR | Avg. |
|---|---|---|---|---|---|
| BASELLM | 82.5 | 39.3 | 58.8 | 38.9 | 54.9 |
| COE | 58.9 | - | 25.9 | - | - |
| SELFDEBUG | 83.3 | 39.3 | 70.6 | 40.0 | 58.3 |
| OPTIMUS | 64.3 | 13.1 | 35.3 | 22.1 | 33.7 |
| VERISIMPL | **91.4** | **42.6** | **70.6** | **45.3** | **62.5** |

*Table 11.* Accuracy of systems, using Mistral as base (except COE).

| | VERISIMPL (Mistral) | |
|---|---|---|
| | Precision | Coverage |
| NL4Opt | 94.9 | 22.8 |
| NLP4LP | 75.0 | 11.5 |
| CompOR | 77.8 | 58.3 |
| IndOR | 87.5 | 16.3 |
| **Avg** | 83.8 | 27.7 |

*Table 12.* Self-verification precision and coverage of VeriSimpl using Mistral.

| System | NL4Opt | | | NLP4LP | | | CompOR | | | IndOR | | | Average | | |
|---|---|---|---|---|---|---|---|---|---|---|---|---|---|---|---|
| | Acc. | Prec. | Cov. | Acc. | Prec. | Cov. | Acc. | Prec. | Cov. | Acc. | Prec. | Cov. | Acc. | Prec. | Cov. |
| VERISIMPL | 88.1 | 92.7 | 32.1 | 51.6 | 90.0 | 28.1 | 76.5 | 100.0 | 53.8 | 45.8 | 83.3 | 22.7 | 65.5 | 91.5 | 34.2 |
| A-CONS | 87.7 | 92.6 | 90.7 | 51.6 | 73.3 | 68.8 | 70.6 | 71.4 | 83.3 | 45.8 | 65.3 | 72.7 | 63.9 | 75.7 | 78.9 |
| A-SINGLEVAR | 88.1 | 92.8 | 86.5 | 46.8 | 76.5 | 44.8 | 76.5 | 76.9 | 76.9 | 47.9 | 62.7 | 69.6 | 64.8 | 77.2 | 69.5 |
| A-FULLVAR | 82.5 | 90.3 | 25.2 | 50.0 | 80.0 | 12.9 | 70.6 | 66.7 | 50.0 | 45.8 | 58.8 | 22.7 | 62.2 | 74.0 | 27.7 |
| A-TYPE | 82.9 | 95.4 | 27.8 | 51.6 | 87.5 | 21.9 | 76.5 | 100.0 | 53.8 | 45.8 | 83.3 | 22.7 | 64.2 | 91.6 | 31.6 |

*Table 13.* Ablation study showing the contribution of different verification components for GPT-4o.

## D. Full Ablation Results

The full ablation results for all systems and benchmarks are shown in Tables 13 and 14.

| System | NL4Opt | | | NLP4LP | | | CompOR | | | IndOR | | | Average | | |
|---|---|---|---|---|---|---|---|---|---|---|---|---|---|---|---|
| | Acc. | Prec. | Cov. | Acc. | Prec. | Cov. | Acc. | Prec. | Cov. | Acc. | Prec. | Cov. | Acc. | Prec. | Cov. |
| VERISIMPL | 88.8 | 95.8 | 38.5 | 57.4 | 75.0 | 17.1 | 76.5 | 66.7 | 15.4 | 67.4 | 76.5 | 20.3 | 72.5 | 78.5 | 22.8 |
| A-CONS | 88.1 | 92.5 | 93.2 | 55.7 | 71.4 | 73.5 | 76.5 | 72.7 | 61.5 | 65.3 | 74.6 | 75.8 | 71.4 | 77.8 | 76.0 |
| A-SINGLEVAR | 88.1 | 93.4 | 95.8 | 55.7 | 66.0 | 91.2 | 70.6 | 73.3 | 91.7 | 61.1 | 68.0 | 94.8 | 68.9 | 75.2 | 93.4 |
| A-FULLVAR | 87.4 | 96.6 | 72.0 | 55.7 | 66.7 | 53.0 | 76.5 | 77.0 | 77.0 | 60.0 | 69.8 | 65.0 | 69.9 | 77.5 | 66.7 |
| A-TYPE | 88.8 | 95.8 | 38.5 | 57.4 | 75.0 | 17.1 | 76.5 | 66.7 | 15.4 | 66.3 | 81.3 | 20.6 | 72.3 | 79.7 | 23.0 |

*Table 14.* Ablation study showing the contribution of different verification components for R1.

# E. Prompts

This section contains the LLM prompts we use for code generation and various forms of verification.

*Figure 4.* Prompt for Gurobi code generation

## Gurobi Code Generation Prompt

System Prompt:

You are an expert in operations research. You specialize in solving Linear Programming (LP) and Mixed-Integer Linear Programming (MILP) problems. Your task is to provide the Gurobi code implementation for the MILP formulation to solve the given problem. Write the Gurobi code exactly as you normally would, but load the input data from a JSON file called data.json. This file will be located in the same directory. At the beginning of the code, define a variable DATA_FILE = "data.json". In addition, load the input JSON data using this variable. Do not hardcode the file name directly inside open(). After loading, proceed to define the variables, objective, and constraints based on the loaded data.

Before starting to write the Gurobi model, carefully understand the problem and think step-by-step about the following: How to model the structure, sets, variables, objective function, and constraints in order to achieve the optimal solution.

The code should be directly executable. Do not include any non-code text, explanations, or formatting like ```python or markdown blocks. Only output clean executable Python code.

Always use the exact variable names and capitalization (uppercase/lowercase) from the data.json file consistently throughout the entire Gurobi code. Do not invent or modify variable names in the Gurobi code.

Do not use any conditional statements (e.g., if) involving Gurobi expressions (LinExpr or QuadExpr) in the logic because these are not directly supported by Gurobi and will raise a NotImplementedError. Instead, model conditional logic using binary variables and additional constraints wherever needed.

IMPORTANT #1: Do not use strict inequalities (< or >) in the model constraints. Always use <=, >=, or = because Gurobi only supports these types of inequalities in mathematical formulations.

If a problem explicitly mentions strict inequality like x > 0 or x > 5, use a small epsilon (e.g., 1e-4) and model it as x >= epsilon or x >= 5 + epsilon. Do not use epsilon unnecessarily. If the problem allows zero production (e.g., x >= 0), model it directly as x >= 0 without epsilon.

IMPORTANT #2: When modeling absolute values, maximums, or minimums of Gurobi expressions, do not use Python functions like abs(), max(), or min(). If the problem requires modeling them, use auxiliary variables and constraints like the following:

To model: m = max(x1, x2):
    model.addConstr(m >= x1)
    model.addConstr(m >= x2)
    (If needed, add m <= threshold)

To model: z = |x1 - x2|:
    model.addConstr(z >= x1 - x2)
    model.addConstr(z >= x2 - x1)

Use these structures only if the problem explicitly states a maximum or absolute value is required.

Here is an example of an input you might receive:

Problem Description:
A company needs to decide how much to produce of two products to maximize revenue. There is a material constraint and a maximum production limit. The goal is to maximize revenue.

Data (data.json):
{
  "NumProducts": 2,
  "Revenue": [50, 70],

```
  "MaterialUsed": [3, 5],
  "MaterialAvailable": 20
}

This would be the correct Gurobi code (a correct output you would produce):

Correct Gurobi Code:
import json
import gurobipy as gp
from gurobipy import GRB

DATA_FILE = 'data.json'
with open(DATA_FILE, 'r') as f:
    data = json.load(f)

NumProducts = data['NumProducts']
Revenue = data['Revenue']
MaterialUsed = data['MaterialUsed']
MaterialAvailable = data['MaterialAvailable']

model = gp.Model()
x = model.addVars(NumProducts, vtype=GRB.CONTINUOUS, lb=0, name='x')

model.setObjective(
    gp.quicksum(Revenue[i] * x[i] for i in range(NumProducts)),
    GRB.MAXIMIZE
)

model.addConstr(
    gp.quicksum(MaterialUsed[i] * x[i] for i in range(NumProducts)) <= MaterialAvailable,
    'MaterialConstraint'
)

model.optimize()

output = {
    'x': [x[i].X for i in range(NumProducts)],
    'TotalProfit': model.ObjVal
}
print(output)

End of Example.

User Prompt Template:

PROBLEM INFO:
{description}

DATA JSON FILE:
{data_json}

You must load each key from data[...] exactly as it appears here without modification. Generate
the Gurobi code based on this information.
```

*Figure 5.* Prompt for Gurobi code error correction

## Gurobi Code Error Correction Prompt

```
System Prompt:

You are a Gurobi Python code corrector. Your task is to correct the Gurobi Python code provided in
the input. The input will contain the standard output and standard error messages from running a
Gurobi Python file. You should analyze these messages and provide a corrected version of the
Gurobi Python code. If there are no errors, you should return the original code. You should return
the corrected code with no further commentary or extra words.

User Prompt Template:

Here is the Gurobi Python code:
{code}

Here is the standard output from running the Gurobi Python file:
{std_out}

Here is the standard error from running the Gurobi Python file:
{std_err}

Please provide the corrected version of the Gurobi Python code. Return only the corrected code,
with no further commentary or extra words.
```

*Figure 6.* Prompt for constraint-based verification

---

**Constraint Verification Prompt**

```
System Prompt:

You are given a problem description with its decision variables. Along with it, several candidate
variable assignments are provided as test cases, and a separate block of problem data containing
all fixed parameters. Your task is to write a python program that evaluates each test case
assignment against all the constraints in the problem and determines whether it is feasible under
the given constraints (please note that we are not concerned with whether an assignment is
optimal, only that it is allowed under the given constraints).

The python program should ensure that the variables are correctly read into appropriate types and
assigned their values as given, compute any derived values, and evaluate each constraint as
boolean checks. You can display reasoning in comments but ensure all aspects of the problem
description (explicit and implicit constraints) are properly tested in the code. Always use the
exact values provided in the assignments with appropriate types and no rounding or approximate
results. The output should only be the executable python code and nothing else (so it can easily
be extracted and executed in python). The final result of the program should be assigned in two
special variables that are a list of strings: passing_tests should contain the labels of all the
test cases that passed and failing_tests should contain the labels of all test cases that failed.
Always make sure that all constraints and calculations operate on the correct data types (e.g.,
use scalar numbers for arithmetic and comparisons, not lists or other containers). If the data
contains nested structures, index properly to extract the needed scalar values before using them.
The output must be ONLY raw Python code that can be executed directly with `python`, without any
extra formatting. Do NOT include Markdown code fences such as ```python or ``` in the output.

Important Instructions:
- You must include ALL test cases provided in the input in the `test_cases` dictionary, without
  skipping, abbreviating, summarizing, or replacing them with comments like "# Add other test
  cases here..." or "# Additional test cases omitted for brevity".
- Do NOT shorten the test_cases dictionary for brevity. The code will be executed automatically
  in a pipeline, so it must be a complete, runnable program. For example if the problem has 28
  Test cases, then all the testcases should be included in the test_case dictionary from E1 to
  E28 and shouldn't be omitted for brevity.
- If there are many test cases, still include them all explicitly in the dictionary exactly as
  given.
- Do not add any explanatory text or markdown formatting (no ``` fences).

A one-shot illustrative example is given below.

Problem Description:
A confectionery company is assembling gift hampers that contain sweets. They can fill the hampers
using candy boxes, toffee jars, or chocolate packs. Each candy box contains 50 sweets and takes
12 minutes to prepare. Each toffee jar contains 150 sweets and takes 5 minutes to prepare. Each
chocolate pack contains 100 sweets and takes 10 minutes to prepare. The company has a total of
400 minutes of preparation time available. To ensure cost efficiency, the company prefers to use
more candy boxes than toffee jars and at least as many chocolate packs as toffee jars. The
objective is to prepare at least 1500 sweets while satisfying all constraints.

Data parameters:
data = {
    "sweets_per_candy": 50,
    "sweets_per_toffee": 150,
    "sweets_per_chocolate": 100,
    "min_total_sweets": 1500,
    "time_per_candy": 12,
    "time_per_toffee": 5,
    "time_per_chocolate": 10,
    "max_total_time": 400
}
```

```
Decision Variables:
- C = Number of Candy Boxes
- T = Number of Toffee Jars
- P = Number of Chocolate Packs

Test Cases:
    - TEST CASE #1, LABEL: <E1>, ASSIGNMENT: C = 10, T = 0, P = 28
    - TEST CASE #2, LABEL: <E2>, ASSIGNMENT: C = 10, T = 2, P = 27
    - TEST CASE #3, LABEL: <E3>, ASSIGNMENT: C = 0, T = 0, P = 0
    - TEST CASE #4, LABEL: <E4>, ASSIGNMENT: C = 0, T = 0, P = 1
    - TEST CASE #5, LABEL: <E1>, ASSIGNMENT: C = 5, T = 5.0001, P = 5

Feasibility Checking Program:
    #Include all test cases
    test_cases = {
        "<E1>": {"C": 10, "T": 0, "P": 28},
        "<E2>": {"C": 10, "T": 2, "P": 27},
        "<E3>": {"C": 0, "T": 0, "P": 0},
        "<E4>": {"C": 0, "T": 0, "P": 1},
        "<E5>": {"C": 5, "T": 5.0001, "P": 5}
    }

    data = {
        "sweets_per_candy": 50,
        "sweets_per_toffee": 150,
        "sweets_per_chocolate": 100,
        "min_total_sweets": 1500,
        "time_per_candy": 12,
        "time_per_toffee": 5,
        "time_per_chocolate": 10,
        "max_total_time": 400
    }

    passing_tests = []
    failing_tests = []

    # Iterate through test cases
    for label, vars in test_cases.items():
        C = vars["C"]
        T = vars["T"]
        P = vars["P"]

        is_feasible = True

        # Constraint: total preparation time <= max_total_time
        total_time = (C * data["time_per_candy"]) + (T * data["time_per_toffee"]) + (P *
        data["time_per_chocolate"])
        if not (total_time <= data["max_total_time"]):
            is_feasible = False

        # Constraint: more candy boxes than toffee jars
        if not (C > T):
            is_feasible = False

        # Constraint: at least as many chocolate packs as toffee jars
        if not (P >= T):
            is_feasible = False

        # Constraint: prepare at least min_total_sweets
        total_sweets = (C * data["sweets_per_candy"]) + (T * data["sweets_per_toffee"]) + (P *
        data["sweets_per_chocolate"])
        if not (total_sweets >= data["min_total_sweets"]):
            is_feasible = False
```

```
        if is_feasible:
            passing_tests.append(label)
        else:
            failing_tests.append(label)

    print("Passing test cases:", passing_tests)
    print("Failing test cases:", failing_tests)

User Prompt Template:

Problem Description:
{description}

Data:
{data}

Decision Variables:
{decision_vars}

Test Cases:
{test_cases}

Feasibility Checking Program:
```

---

*Figure 7.* Prompt for variable-based verification

## Decision Variable Verification Prompt

System Prompt:

You are an expert in mathematical optimization and decision modeling. You are verifying the internal consistency of a solved linear programming (LP) problem by inferring optimal values of some missing decision variables.

You will be given:
  - A problem description that defines the optimization goal and the relationships among decision variables.
  - The list of decision variables and their definitions.
  - The known numeric values for some decision variables.
  - The names of missing variables to infer.

Your goal:
  - Use logical and quantitative reasoning to deduce the missing variables' optimal values based on the relationships described in the optimization problem.
  - Perform explicit arithmetic reasoning; use exact numeric values without rounding intermediate results.
  - Reference the problem text naturally in your explanation (e.g., "as stated in the problem...").

Output format -- absolutely mandatory:
  - Return one single JSON object (no Markdown fences, no commentary).
  - The JSON must contain exactly two fields:
      - "reasoning": a short, clear explanation of how you derived the optimal values of the missing variables.
      - "predicted_values": map of missing variable names to inferred optimal values.

Here is an example of what is expected from you.

Problem Description:
A furniture manufacturer produces three products: chairs, tables, and benches. Each chair requires 2 hours of carpentry time and 3 kilograms of wood. Each table requires 4 hours of carpentry time and 5 kilograms of wood. Each bench requires 3 hours of carpentry time and 4 kilograms of wood. The company has a maximum of 60 labor hours and 70 kilograms of wood available per week. The profit earned per unit is $20 for a chair, $35 for a table, and $25 for a bench.

The manufacturer wants to **maximize total profit**, subject to the labor and wood constraints.

Decision Variables:
    - x = number of chairs produced
    - y = number of tables produced
    - z = number of benches produced

Known Values:
    x = 5

Missing Variable to Predict:
    y
    z

Output:
{
"reasoning": "The company can use up to 60 hours of carpentry time and 70 kilograms of wood each week. Since each chair takes 2 hours and 3 kilograms of wood, producing 5 chairs uses 10 hours and 15 kilograms of wood. That leaves 50 hours of labor time and 55 kilograms of wood available for making tables and benches. Each table needs 4 hours and 5 kilograms of wood, while each bench needs 3 hours and 4 kilograms. The total weekly profit of 457 dollars includes the profit from the 5 chairs plus whatever is earned from the tables and benches. To reach that level of profit while staying within the available wood and labor, the manufacturer would likely make about 8 tables and 3 benches. This combination fits comfortably within the remaining time and wood limits

```
and gives a total profit close to the reported value.",
"predicted_values": {"y": 8, "z": 3}
}

User Prompt Template:

Given the following linear programming problem, predict the numeric value of the missing decision
variable using the problem description and the known variable values.

When performing calculations:
  - Use exact numeric values and do not round intermediate steps.
  - Round the final answer only if needed (two decimal places max).
  - Maintain consistency with the quantitative relationships stated in the problem.

Problem Description:
{description}

Decision Variables:
{decision_vars}

Decision Variable Definitions:
{decision_var_definitions}

Known Values:
{known_values}

Missing Variable to Predict:
{target_var}

Instructions:
  - Use the relationships described in the problem to infer the value of {target_var}.
  - Show substitutions and calculations explicitly.
  - Reference the problem text naturally (e.g., "as stated in the problem...").
  - Return only a single JSON object with exactly two fields:
      - "predicted_value": <float>
      - "reasoning": "<brief explanation>"
```

*Figure 8.* Prompt for type-based verification

---

**Type Verification Prompt**

```
System Prompt:

Task:
You are given a real-world word problem and data involving a linear programming or optimization
scenario, and the Gurobi code where the decision variables have unknown types. Please reason
about what each of the decision variables represent and what their appropriate types should be.

Your job is to:
  - Understand what the decision variables (DVs) mean in the problem.
  - Decide the appropriate Gurobi variable type for each DV, based strictly on:
      * The problem wording
      * The real-world logic (can the quantity be continuous? is it countable? is it binary?)
    Explain your reasoning based on:
      * Whether the item is countable or discrete (like jars, trips, meals, vaccines -> use
        integer)
      * Or whether the item is divisible or measurable (like grams, hours, medicine doses ->
        use continuous)

Output:
Return a valid JSON object. Each decision variable name is a key. The value for each key is a
three-element list:
  [meaning, reasoning, final_answer]
where
  - meaning  = explanation of what the DV represents and means in the problem
  - reasoning = explanation of why that variable's type was chosen
  - final_answer = one of the following labels: "integer", "continuous", or "binary"

Output only the JSON. No extra text or explanation.

IMPORTANT NOTE:
Ensure the number of keys in the JSON you return is equal to the number of decision variables in
the code. This means ensure the number of keys in the output JSON equals the number of times
.addVar(...) appears.

Problem Example:

Description (from input):
A gardener must give his vegetable patch at least 1000 litres of water today while keeping his
electricity use as low as possible. He can run a garden hose, which delivers water in whatever
exact amount he lets flow but costs 0.05 kWh of electricity for every litre the pump pushes
through. He can roll out ready-filled 100-litre barrels, each of which costs 4 kWh to hoist and
tip into the beds. And he can switch on a sprinkler system that soaks the plot with 500 litres
in a single run and draws a flat 60 kWh, though regulations allow him to use that sprinkler no
more than once in the day. The task is to choose how much water to supply by hose, how many
barrels to empty, and whether to activate the sprinkler so that the total volume reaches or
exceeds 1000 litres while the combined electricity consumption from these choices is minimised.

Data (from input):
{
    "n_water_options": 3,
    "water_per_unit_litres": [1, 100, 500],
    "electricity_per_unit_kwh": [0.05, 4, 60],
    "min_water_requirement_litres": 1000,
    "max_sprinkler_runs": 1
}

Code (from input):
import gurobipy as gp
from gurobipy import GRB

m = gp.Model("garden_watering")
```

```
x_H = m.addVar(lb=0)
x_B = m.addVar(lb=0)
x_S = m.addVar()

m.addConstr(x_H + 100 * x_B + 500 * x_S >= 1000, name="min_water")

energy = 0.05 * x_H + 4 * x_B + 60 * x_S
m.setObjective(energy, GRB.MINIMIZE)

m.optimize()

if m.status == GRB.OPTIMAL:
    print("Optimal watering plan")
    print(f"  Hose water (L)      : {x_H.X:.1f}")
    print(f"  Barrels used        : {int(x_B.X)}")
    print(f"  Sprinkler on (0/1)  : {int(x_S.X)}")
    print(f"  Total electricity   : {energy.getValue():.2f} kWh")
else:
    print("No feasible solution found.")

Sample Output for Example:
{
    "x_H": [
        "Amount of water (in litres) delivered using the garden hose; this is clear from the
        name 'x_H' and from the constraint where it's added directly, indicating it contributes
        to the total water volume, and the cost per litre is 0.05 kWh.",
        "Water delivered through a hose can be controlled very precisely, allowing the gardener
        to choose any amount down to fractions of a litre. Since the hose is not constrained to
        fixed units and the electricity cost is per litre, it logically follows that partial
        litres are both possible and meaningful in this context. Therefore, the variable can
        allow non-integer values to reflect real-world precision.",
        "continuous"
    ],
    "x_B": [
        "Number of 100-litre barrels used; 'x_B' appears in the model as being multiplied by
        100 to give litres of water, and each unit consumes 4 kWh, indicating it models discrete
        barrels.",
        "Barrels are pre-filled and fixed in size (100 litres), so the gardener can only use a
        whole barrel at a time. It would not make practical sense to use part of a barrel in
        this context, as that would require an artificial way to divide a barrel physically.
        The decision revolves around how many full barrels to use, meaning the values must be
        whole numbers.",
        "integer"
    ],
    "x_S": [
        "Indicator for whether the sprinkler system is activated; it contributes a flat 500
        litres and 60 kWh if used, and the problem mentions that it can be used at most once.",
        "The sprinkler system is either used or not used, with no possibility of partial
        activation. It provides a fixed volume and consumes a fixed amount of energy in one
        operation. This on/off nature is a classic binary decision--using it means setting the
        variable to 1, and not using it means setting it to 0.",
        "binary"
    ]
}

User Prompt Template:

Tell me the type of this problem given the following:

Description:
{description}
```

```
Data:
{data}

Gurobi Code:
{masked_code}

Ensure the number of keys in the JSON you return is equal to the number of decision variables in
the code. This means ensure the number of keys in output JSON equals number of times .addVar(...)
appears. If the decision variable is represented as an array, only return 1 key which would be
the name of the variable given in the code above.
```

