# OpenReview forum: "VeriSimpl: Robust Optimization Modeling from Natural Language using Simplification-based Verification"
_ICML.cc/2026/Conference — ICML 2026 regular_

### Official Review · Reviewer_qLjZ · 2026-02-16

**Soundness:** 3
**Presentation:** 2
**Significance:** 2
**Originality:** 3
**Overall Recommendation:** 4
**Confidence:** 4

**Summary:**

This paper addresses the research question about how to generate better solver formulation from natural language input utilizing LLMs. The authors argue that a big problem for these modeling tasks is that the inferred solver formulation is not representing the intended tasks correctly. They try to solve this research question by introducing two types of verifications: contraint-based verification and variable-based verification, both using a pipeline solver -> LLM. Their experiment results on different datasets demonstrate improvements on formulation accuracy compared to previous benchmarks.

**Compliance With Llm Reviewing Policy:**

Affirmed.

**Final Justification:**

The rebuttal period addressed my main concerns about this paper.

**Key Questions For Authors:**

1. How to calculate $s_c, s_d, s_f, s_t$ is not explicit to me in Algorithm 1. Could you explain it in more details?

2. For constraint-based verification, is it the case that for each constraint, only one value for each of the mutations (<, =, >) are generated and provided to LLM for verification, or is it that multiple values are sampled for each of the mutations?

**Limitations:**

yes

**Strengths And Weaknesses:**

Strengths:
1. The proposed verification methods leverage solvers to simplify the process, which seems sound.
2. Experiment results show that $VeriSimpl$ improves the formulation accuracy.

Weaknesses:
1. The paper cannot solve the case where constraints are missed by LLMs when doing formulation.
2. It is a bit redundant between section 2 and section 3 as they both discuss the verification methods. I encourage the author to adjust writing.

---

> ### Author Rebuttal · Authors · 2026-03-31
>
> Many thanks to the reviewer for the very helpful and constructive feedback.
>
> **W1**. We agree that detecting missing constraints is an important direction for future work (as also discussed in section 4.4). One possible extension is to incorporate coverage-oriented verification checks, where the system explicitly checks whether all key elements of the natural language description (e.g., entities, constraints, etc) are represented in the formulation, potentially using LLM-based extraction or alignment techniques. Another potential direction is to combine the current approach with additional test cases that are inferred directly from the NL description and tested using the solver – which will be a *bidirectional* kind of verification approach where some tests are generated using the solver and verified by LLM, while others are generated by the LLM and checked with the solver.  We will add these as interesting directions for future work.
>
> **W2**. Thank you and we agree that there is some redundancy between Sections 2 and 3. Our intention was for Section 2 to provide a high-level conceptual overview of the simplification-based verification approach, and Section 3 to present the formal algorithmic details. We will revise the paper to make this distinction clearer and avoid overlap in the descriptions.
>
> **Q1**. In Algorithm 1, the scores correspond to different verification components that provide verification signals from different aspects:  s_c  is the constraint-based verification score (Algorithm 2), computed as the proportion of feasibility queries that pass (each feasibility query considers mutations of one constraint while keeping others fixed, and asks the LLM to reason about the mutations in isolation). The scores  s_d  and s_f  are variable-based verification scores (Algorithm 3), computed as the proportion of valuation queries that pass (each valuation query fixes given values for certain variables and asks the LLM to infer optimal values for the remaining masked variables). s_d  is the single-variable masking score (one variable masked while all others given) and s_f  is the full variable masking score (all variables masked). Finally,  s_t  is the type-consistency score, which is a binary indicator returned by the verification of variable types by the LLM. These scores are combined as a tuple (s_c + s_d + s_f, s_t) and compared lexicographically (semantic verification scores followed by syntactic type-verification score) to select the best candidate program. We will improve the writeup to clarify these definitions in the paper.
>
> **Q2**.
> In our current implementation, for each constraint and each mutation type we generate a single witness valuation using the solver, which is then used to construct the corresponding feasibility query for the LLM. We chose this for efficiency, as it keeps the number of verification queries manageable while still providing systematic coverage of the logical possibilities in the constraint space. However, the framework naturally supports generating multiple witness valuations per mutation, which could further improve robustness by probing a wider range of behaviors (at higher computational cost). We will clarify these design possibilities/trade-offs in the paper and discuss this extension as future work.

---

> > ### Author Rebuttal · Reviewer_qLjZ · 2026-04-02
> >
> > Thank you for the detailed response. I'll maintain my positive score 4.

---

### Official Review · Reviewer_ZFQp · 2026-03-10

**Soundness:** 3
**Presentation:** 3
**Significance:** 3
**Originality:** 3
**Overall Recommendation:** 4
**Confidence:** 3

**Summary:**

This paper introduces VeriSimpl, a solver-LLM framework for translating natural language descriptions into executable optimization solver formulations. The key contribution is a paradigm called simplification-based verification, which inverts the conventional verification workflow: rather than having the LLM generate test cases for the solver to check, the optimization solver is used to generate simplified diagnostic queries that the LLM can tractably reason about. Concretely, the framework generates several candidate programs via LLM sampling, then applies three complementary verification signals (constraint-based, variable-based, and type-based) to score and select the best candidate. Evaluations across four benchmark datasets with three base LLMs show consistent accuracy improvements over existing baselines, and the self-verification signal achieves high precision (>90% for GPT-4o on average) when all verification checks pass, albeit with low coverage (20–34%).

**Compliance With Llm Reviewing Policy:**

Affirmed.

**Final Justification:**

I thank the authors for their detailed rebuttal. The response addressed most of my concerns, especially by improving the transparency of the experimental setup, clarifying the handling of solver failures and variable matching, and correcting the reporting issues in the tables.

Overall, my assessment remains positive. I believe the paper makes a solid contribution, with a well-motivated idea and consistent empirical gains across models and benchmarks. The rebuttal reinforced my prior assessment, and I will maintain my current score.

**Key Questions For Authors:**

- **Q1. Computational cost and fairness of comparison.**

VeriSimpl generates K=10 candidate programs and applies multiple LLM verification calls per candidate. What is the total number of LLM calls (or approximate token cost) incurred by VeriSimpl compared to each baseline in a typical problem instance? Were baselines such as SelfDebug also configured with comparable sampling budgets? If not, have the authors considered a cost-controlled comparison, such as fixing the total LLM call budget and comparing VeriSimpl against Best-of-N selection with a simpler scoring heuristic?

**Q2. Numerical tolerance in variable-based verification.**

For continuous decision variables, how is correctness defined in the variable masking verification (Algorithm 3, \hat{v}t = v*t)? Is exact floating-point equality required, or is a tolerance threshold applied? If a tolerance is used, how was it chosen?

**Q3. Solver failure rate under constraint mutation.**

How often does the solver fail to return a feasible or optimal solution for mutated problem instances during constraint verification? Does this rate vary systematically across benchmark datasets or problem types? And does the silent skipping of these cases introduce any detectable bias in the verification scores?

**Q4. Iterative refinement using verification feedback.**

The current framework uses verification scores only for candidate selection. Have the authors considered using the specific verification failures (e.g., which constraints failed feasibility checks, which masked variables were incorrectly predicted) as targeted feedback for iterative LLM refinement of the candidate program? What are the practical obstacles to this extension, and would it be a natural direction for future work?

**Q5. Experimental reporting details.**

Could the authors clarify the experimental setup for the IndOR dataset, where reported accuracy values include fractional percentages despite the dataset containing 100 integer-count samples? Were experiments repeated across multiple runs, and if so, what variance was observed? Additionally, could the authors justify the use of simple (unweighted) average accuracy across datasets of substantially different sizes?

**Limitations:**

Yes.

**Strengths And Weaknesses:**

### Strengths

- S1. **Consistent improvements across models and benchmarks.** Results hold across three base LLMs and four datasets spanning different problem domains and difficulty levels, suggesting the approach is not overfitted to a specific model or benchmark distribution.

- S2. **Complementary verification signals with empirical support.** The three verification components address distinct aspects of formulation correctness, i.e., feasibility, optimality, and type consistency. The ablation study confirms that each contributes meaningfully. No single signal dominates across all benchmarks, supporting the value of their combination.

- S3. **Honest failure analysis.** The appendix includes detailed failure cases that clearly articulate the structural limitations of the approach, particularly the inability to detect errors in decision variable definitions and missing problem aspects.

### Weaknesses

- W1. **Computational cost comparison is not controlled.** Verisimpl incurs substantially higher computational cost than baselines: K=10 candidate programs are generated, and each undergoes multiple LLM calls for constraint, variable, and type verification. The baselines, particularly BaseLLM and SelfDebug, are not evaluated under equivalent computational budgets (e.g., asking LLM to generate multiple solver formulations and pick one as the final answer). It is therefore unclear how much of the performance gain is attributable to the verification signal itself versus the increased inference-time computation. A fair comparison would require either fixing the total LLM call budget across systems, or explicitly reporting and comparing cost-normalized performance.

- W2. **Solver reliability under constraint mutation is not discussed.** The constraint verification procedure relies on the solver successfully returning `OPT` or `FEAS` status for each mutated problem instance. However, mutations may produce numerically ill-conditioned problems, causing the solver to fail (e.g., `TIME_LIMIT`, `ITERATION_LIMIT`, `NUMERIC` by Gurobi); while these cases are simply skipped (Algorithm 2, line 6). The paper does not report how frequently this occurs, nor does it discuss whether systematic skipping of certain constraint types could introduce bias into the verification signal.

- W3. **Experimental reporting lacks transparency.** The accuracy values for IndOR (100 samples) include fractional percentages (e.g., 42.7%), which is inconsistent with integer sample counts unless results are averaged over multiple runs, but no information about random seeds, number of runs, or variance is provided. Additionally, the average accuracy across benchmarks is computed as a simple mean over datasets of very different sizes (17 to 269 samples), which may misrepresent overall system performance. A weighted average, or at minimum an explicit justification of the aggregation choice, would be more appropriate.

- W4. **Structural false negatives undermine self-verification reliability.** The self-verification signal suffers from structurally inevitable false negatives. The LLM used as a verifier is itself imperfect: it may misjudge feasibility queries due to ambiguous problem descriptions or numerical reasoning errors, and variable-based verification requires exact numerical matching against solver outputs, which is particularly problematic for continuous variables where LLM-inferred values will rarely match floating-point solver results exactly. The paper acknowledges low coverage (20–34%) but does not analyze the sources of false negatives, nor does it discuss whether relaxed matching criteria could improve coverage without sacrificing precision.

---

> ### Author Rebuttal · Authors · 2026-03-31
>
> Many thanks to the reviewer for the very helpful and constructive feedback.
>
> **W1/Q1**. Thank you for raising this point. Please see our response to **Reviewer twEV  under Q1 (*Cost-controlled comparison*)**.
>
> **W2/Q3**. We reran our experiments to infer the percentage of mutation cases where a feasible solution is not found. Overall, this happens in 18.5% of cases and this rate does not vary significantly over different benchmarks:  NL4Opt (14.7%), NLP4LP (19.2%), CompOR (23.0%), IndOR (16.9%). We agree this is an area for further improvement. It can happen when there is overlap between constraints: e.g. if we fix all constraints apart from C and they overlap with C, then that prevents a feasible instantiation of the negation of C, and hence violations of C are not checked. Hence this is another source of incomplete coverage (similar to missed constraints discussed in Section 4.4). Future work could address this by grouping and verifying overlapping constraints jointly and we will add this discussion.
>
> **W3/Q5**. We apologize for an error in the write-up: as with the NL4Opt dataset, we had also identified and removed some infeasible cases from IndOR and NLP4LP, but mistakenly reported the original dataset sizes. The correct counts are 96 problems for IndOR (after removing 4) and 62 for NLP4LP (removed 5). We will correct this mistake in the writeup. Experiments were not run multiple times (API cost constraints and  also common practice in related works) and we also did not find noticeable variability/inconsistency in repeat runs during development as our method already does internal sampling of many programs.
>
> While the correct dataset counts explain the fractional results for IndOR, we also detected some minor errors in the original percentage calculations in some cells in Tables 1 and 2  that we clarify below. Note that these corrections do not change any conclusions of the evaluation  as the differences are very small: overall averages change by  less than 0.3%  for all systems except Optimus which changes by 1.1% (but remains the lowest performing system). VeriSimpl remains the best-performing method across all benchmarks and the relative ranking of all methods is unchanged. The actual revisions to Table 1 and 2 are:
>
> *Table 1 (accuracies with GPT-4o as base):*
>
> |System| NL4Opt| NLP4LP|CompOR|IndOR|Avg.|
> |-|-|-|-|-|-|
> |BaseLLM|75.5|~~44.3~~ → 43.5 |64.7|42.7|~~56.8~~ → 56.6|
> |CoE|58.9|-|25.9|-|-|
> |SelfDebug|76.5|46.8|70.6|42.7| 59.1|
> |Optimus| 85.9 |~~24.0~~ → 37.1 |52.9| ~~26.6~~ → 17.7|~~47.3~~ → 48.4|
> |VeriSimpl|88.1|51.6|76.5|45.8|65.5|
>
>
> *Table 2  (accuracies with R1 as base):*
> |System|NL4Opt|NLP4LP|CompOR|IndOR | Avg.|
> |-|-|-|-|-|-|
> |BaseLLM|78.4| ~~50.8~~ → 51.6 | 64.7| ~~55.8~~ →55.2 |~~62.4~~ → 62.5|
> |CoE|58.9|-|25.9|-|-|
> |SelfDebug|82.9|~~55.7~~ → 56.5|70.6| ~~62.1~~ → 61.5 |~~67.8~~ → 67.9|
> |Optimus|84.4| ~~31.1~~ → 32.2|47.1|~~56.8~~ → 56.3 |~~54.9~~ → 55.0|
> |VeriSimpl|88.8| ~~57.4~~ → 58.1|76.5| ~~67.4~~ → 67.7|~~72.5~~ → 72.8|
>
> We will update the Tables in the paper with these revisions. Regarding aggregation, we used an unweighted average to treat each benchmark equally, as they represent different problem domains and sources. But  we agree a weighted average  is also informative and have computed them as follows:
>
> |System|Avg (GPT4o)|Weighted Avg (GPT4o)|Avg (R1)|Weighted Avg (R1)|
> |-|-|-|-|-|
> |BaseLLM|56.6|63.5|62.5|69.1|
> |SelfDebug|59.1|64.8|67.9|74.1|
> |Optimus|48.4|63.1|55.0|69.6|
> |VeriSimpl|65.5|73.4|72.8|79.5|
>
> Similar conclusions can be seen for weighted averages: VeriSimpl remains the best-performing method across all benchmarks, with in fact even larger gains over SelfDebug and BaseLLM. Interestingly, Optimus sees a significant boost with weighted average as it performed best on the largest NL4Opt dataset (though it is still lower than the other SelfDebug baseline).
>
> **W4/Q2**. We agree that false negatives cause low verification coverage as we also discuss in section 4.2. They are mainly due to numerical reasoning errors by the LLM, overlooking certain constraints or ambiguity in the problem description. For variable equality matching, we do use a small error tolerance (ε = 1e−4)  rather than exact matching – this tolerance was chosen mainly to account for floating-point discrepancies. In our experiments we found that larger differences in values were commonly due to  true semantic differences rather than approximation errors (real differences between LLM reasoning and program outputs). Hence further relaxation of matching may degrade accuracy, though this does merit further investigation.
>
> **Q4**. Yes this is a very good direction for future work. We are already refining programs based on types verification, and extending to semantic verification failures will be valuable. A key challenge is reliably determining whether failures stem from LLM reasoning errors or actual program issues, as acting on false negatives could lead to incorrect modifications of correct programs.

---

> > ### Author Rebuttal · Reviewer_ZFQp · 2026-04-03
> >
> > Thank you to the authors for the detailed clarifications. The rebuttal addresses most of my concerns and improves the transparency of the experimental setup. I will maintain my current score.

---

### Official Review · Reviewer_2XLo · 2026-03-12

**Soundness:** 3
**Presentation:** 2
**Significance:** 1
**Originality:** 3
**Overall Recommendation:** 4
**Confidence:** 4

**Summary:**

This paper studies verification for natural-language-to-optimization generation. The main idea is to avoid checking a candidate formulation only at the full-problem level, and instead use the solver to generate simplified diagnostic queries that probe whether the candidate program matches the natural language specification on more localized reasoning steps. The method combines constraint-based, variable-based, and type-based checks to score candidate programs, and the experiments show improvements over baseline generation systems across several benchmarks.

**Compliance With Llm Reviewing Policy:**

Affirmed.

**Key Questions For Authors:**

1.	Can the authors give a clearer high-level explanation of the full verification pipeline, especially the intuition behind the constraint-based and variable-based simplifications, and how they differ?
2.	How should readers interpret the low verification coverage in practice? Is the intended use case full automation, or mainly reducing manual verification effort?
3.	The ablations suggest the full method helps, but the gains seem somewhat modest. Where do the authors think the proposed verification scheme matters most?
4.	Can the authors clarify the main failure modes, especially cases where the verification process may reject correct formulations or still accept incorrect ones?

**Limitations:**

Partially. The authors acknowledge some limitations in Section 4 and the appendix, but the low-coverage issue and the failure modes illustrated in Appendix B deserve more explicit treatment in the main paper. There are no significant negative societal impacts beyond standard considerations for AI-assisted optimization tools.

**Strengths And Weaknesses:**

I think the paper tackles an important problem. Generating optimization models from natural language is difficult precisely because many wrong formulations can look locally plausible while being semantically wrong, so the general direction here makes sense. The core idea of probing the model’s understanding through simplified solver-grounded queries is interesting, and, at least to me, feels like a nontrivial contribution. Even though I am not deeply familiar with the optimization-specific literature, this does feel meaningfully different from just sampling candidates and picking one by surface heuristics. Empirically, the method appears to improve over prior systems, which is a real strength.


My main issue is that the paper is harder to follow than it should be, especially because the contribution is primarily methodological. I could usually follow the local pieces in isolation, but I often did not come away with a clean end-to-end picture. Several specific points caused persistent confusion: the notation in Figure 2 (the P and V nodes) is never explained, making the pipeline diagram hard to parse; Algorithm 1 lines 4–8 introduce four score components without explaining how they interact or how ties are broken; and the variable-based simplification asks the LLM to infer optimal variable values given all others fixed, which raises the question of why this is more reliable than LLM generation on the original problem — a question the paper does not directly address. The paper needs a clearer intuitive explanation of these elements before the formal presentation.

I also think the practical impact is somewhat mixed in its current form. On the positive side, the empirical results do suggest that the proposed verification scheme is useful: the method seems to improve over baseline systems, and the general idea of probing whether the model “understands” the generated formulation is interesting. But the results also suggest that the approach is still fairly limited as a robust verification layer. In particular, the verification coverage is low, which makes the method feel somewhat restrictive, and the ablations indicate that the gains from the full pipeline, while real, are not especially large. So my overall take is positive but qualified: I do think there is a meaningful and interesting idea here, and it seems better than just relying on end-to-end generation alone, but the paper does not yet fully convince me that the proposed framework is mature or broadly effective in practice.

---

> ### Author Rebuttal · Authors · 2026-03-31
>
> Many thanks to the reviewer for the very helpful and constructive feedback.
>
> **Weaknesses (Specific clarifications)**. We will revise the presentation to improve clarity on these points:
>
> - In Figure 2, the P nodes are candidate programs (e.g. Gurobi code) and V nodes are verification queries derived from solver-based simplifications (Feasibility and Valuation queries formally defined in Section 3).
> - In Algorithm 1, the score components include 3 semantic verification signals  (constraints, single-variable and full-variable) and the one syntactic type verification score. These scores are computed independently as separate normalized signals examining different program properties, and the 3 semantic scores are then aggregated into a single score (equal weighting) with ties broken by the syntactic type-based score (lexical ordering in line 8).
> - Variable-based simplification improves reliability by reducing problem dimensionality. By fixing most variables to solver-derived values, the LLM only needs to infer a small subset, rather than jointly solving for all variables. For example, instead of inferring mutually optimal values for x, y, z together, the task becomes: given x = v1 and y = v2, what is the optimal value of z? This localizes reasoning to a low-dimensional subproblem, avoiding complex global inference.
>
> **Q1.**  At a high level, our method generates candidate programs and evaluates them using  solver-generated simplified queries that probe localized properties, reducing global reasoning into tractable subproblems. This is done in two main ways:
> - Constraint-based verification: for each constraint, assuming all others fixed, we test whether it is correctly implemented by the program  by generating concrete satisfying/violating instances and asking the LLM to reason about them.
> - Variable-based verification:  given concrete values for most variables, we ask the LLM to infer the remaining ones, reducing dimensionality (as in W1 response above).
>
> The constraint and variable checks provide normalized verification scores reflecting the number of passing checks, which are aggregated and highest scoring programs selected. We hope this clarifies the pipeline, and are happy to answer any further questions.
>
> **Q2**. In the paper we report coverage for *full verification*, which yields high precision (e.g. >90%) when all checks pass but with limited coverage, and is thus intended as a high-confidence signal rather than a full replacement for generation. However, we also note that our  verification is *not inherently binary*:  it produces a graded score based on how many checks pass, which allows confidence estimation and *prioritization* of manual review in practice (e.g., flagging high-risk cases). Furthermore, our queries also provide fine-grained diagnostic probing (e.g., identifying faulty constraints or objectives), aiding targeted debugging unlike prior methods. Finally, even if not explicitly exposed to users, verification contributes to the overall accuracy gains over baselines, and improving its coverage remains an important direction for future work.
>
> **Q3**. The ablation results show that each component (constraint-based and variable-based verification) contributes significantly on its own, so their gains are highly overlapping. This is expected due to the interconnected nature of optimization problems—for example, an error in a constraint may result in incorrect variable values that may be detected by variable verification even if constraint verification is not explicitly performed. The full combination, however, provides a more complete semantic check, jointly covering feasibility and optimality. This is where our method is most beneficial: detecting subtle semantic errors (e.g., slightly incorrect constraints or objectives) that do not cause execution failures but still lead to incorrect solutions, and are difficult to catch with existing approaches based on execution or syntax alone.
>
> **Q4** False positives and negatives:
> - Verification can reject correct answers mainly due to the strict consistency requirements: even when a program is correct, verification may fail if the LLM produces inconsistent answers to simplified queries e.g., due to numerical reasoning mistakes, overlooking certain constraints or ambiguity in the NL problem (see also section 4.2).
> - Verification may accept wrong answers when both the program and the LLM reasoning share the same incorrect interpretation of the problem. As in Appendix B, this can happen when, for example, a decision variable is misinterpreted  (e.g., “start time” in shift scheduling), leading to solver outputs aligning with the LLM’s reasoning, or when both overlook certain constraints/costs in a profit objective calculation. These cases highlight important directions for future work, e.g. expanding verification to include  decision variable definitions and coverage checks of the NL spec (see also Appendix B and Section 4.4 for more details).

---

> > ### Author Rebuttal · Reviewer_2XLo · 2026-04-03
> >
> > Thanks for the clarification. I have two remaining concerns.
> >
> > First, on the scoring scheme: the rebuttal explains how the four components are combined, but I am still unclear on what makes this aggregation generally well-justified rather than a task-specific design choice. In particular, why should the normalized scores be comparable across different kinds of optimization problems, and what supports the use of equal weighting plus lexicographic tie-breaking as a generally applicable strategy? I think the paper would benefit from clarifying whether this is intended as a principled, domain-agnostic scoring rule or simply a reasonable heuristic that worked well empirically.
> >
> > Second, the discussion of false positives and false negatives makes me less convinced by the framing as “verification.” If the method can reject correct formulations because of LLM inconsistency, and accept incorrect ones when the program and the LLM share the same misunderstanding of the natural language specification, then the method seems closer to a diagnostic or confidence-based checking procedure than verification in the stronger sense. I would encourage the authors to clarify this framing and avoid overstating what guarantees the method actually provides.

---

> > > ### Author Response · Authors · 2026-04-05
> > >
> > > Thank you very much to the reviewer for their response, and we address their additional questions below.
> > >
> > > **Scoring scheme**
> > >
> > > Our scoring scheme is chosen as an equal-weighted aggregation of the semantic verification signals, followed by  types-checking as a secondary signal. Intuitively, it combines the scores from the different semantic verification signals (constraints and variables based verification) as different *features* with equal weights, which is why we normalize each of these signals between 0 and 1. For example, the constraints score s_c represents the degree of success in constraints-based verification (1 means constraints verification has fully passed, 0 means complete failure, 0.7 reflects 70% success). Hence, each of these scores represents the success rate of a different feature of correctness (how many of the verification queries for this feature have passed?), and normalized scores  equally represent the different aspects. In particular, note that we are not comparing the normalized scores across different optimization problems – for a given optimization problem, we normalize *within each kind of verification* and then aggregation yields overall verification success.
> > >
> > > The lexicographic tie-breaking gives priority to semantic correctness over type checking in order to  implement types-verification as  a *secondary* check after the main semantic verification. Intuitively, whether variables in the program have exactly the right types (int or float, etc) is a secondary concern after checking that the main logic of the program is correct -  hence types checking is not given equal weight with the semantic correctness  but is a secondary signal to distinguish between programs that may otherwise pass logical formulation checks.
> > >
> > > We agree that these are simplified design choices that we have adopted in this work. A more comprehensive exploration of this aspect may investigate different aggregation models as well as learning weights from data – which will also require larger training datasets and further experimentation. However, our goal in this work was to show the value of  our solver-guided self-verification approach, which is effectively demonstrated by the simple aggregation model used here. Importantly, the framework is modular: alternative weighting schemes or learned aggregation strategies could be incorporated without changing the underlying verification mechanisms. We will include discussion of these design choices and their further investigation as future directions.
> > >
> > > **Verification framing**
> > >
> > > Thank you for raising this point and we agree this is an important clarification to make. We use verification in the sense of *self-verification* with LLMs (which has become an active research area in LLM reasoning – see [1,2] and related works section) rather than *formal verification* in the sense of formal proofs and guarantees. We firstly note that formal verification for problem settings such as ours is not possible because the problem descriptions are given in natural language, which is itself informal and can always have ambiguity (hence no method can provide correctness guarantees for arbitrary problems described in NL) - that is why we highlight the *high precision* of our approach rather than absolute guarantees. Our notion of verification is based on self-verification by LLMs – where the LLM explicitly reasons about the correctness of its own solutions. Our key novelty is that since direct self-verification by LLMs is extremely challenging for complex high-dimensional optimization problems, we leverage the solver to generate simplified queries that probe different localized aspects of the problem to enable tractable reasoning by the LLM. In this regard, our verification is still based on logical reasoning conducted by the LLM and is therefore different from statistical confidence or uncertainty estimation techniques in ML.
> > >
> > > Finally, while it is not possible to provide full correctness guarantees in our setting, we can provide a theoretical *soundness*  property for our verification queries: if a program is correct (it correctly implements the intended user problem) then our verification queries (both feasibility and valuation queries) must be satisfied. Hence potential errors or uncertainty do not come from the queries themselves, but in the LLM’s ability to reason about them. In contrast, if the LLM were to generate test cases, then the tests themselves may contain errors or uncertainty. Thus our method separates verification query construction (solver-driven, deterministic) from verification evaluation (LLM reasoning, imperfect), and the former is sound under correct specifications.  We hope this clarifies our scope and positioning, and we will also clarify these points in the paper.
> > >
> > > [1] Teaching large language models to self-debug. Chen et al,  ICLR, 2024.
> > >
> > > [2] Large Language Models are Better Reasoners with Self-Verification. Weng et al, EMNLP findings, 2023

---

### Official Review · Reviewer_twEV · 2026-03-18

**Soundness:** 3
**Presentation:** 3
**Significance:** 2
**Originality:** 3
**Overall Recommendation:** 4
**Confidence:** 4

**Summary:**

This paper introduces a novel framework that addresses the challenge of subtle semantic errors in LLM-generated optimization code. Unlike prior methods that rely on LLM-generated tests, VeriSimpl leverages an optimization solver to construct simplified diagnostic queries (via constraint mutation and variable masking, etc). This allows the LLM to perform tractable, localized reasoning to verify if the code’s logic aligns with the natural language intent. Its key contributions include a robust simplification-based verification paradigm that may reduce the manual inspection burden in industrial modeling.

**Compliance With Llm Reviewing Policy:**

Affirmed.

**Key Questions For Authors:**

1. The proposed verification process involves multiple mutation queries per constraint and masking queries per variable, leading to significantly higher Token consumption and latency than baselines. Have you explored adaptive verification strategies—such as prioritizing "risky" constraints or using a cheaper LLM for the simplified reasoning tasks—to make the framework more cost-effective for large-scale industrial problems? Meanwhile, are there any comparative experiments that evaluate inference accuracy against other test-time scaling frameworks under roughly equivalent token consumption?
2. The examples provided (like timber storage) involve relatively intuitive linear relationships. For complex or abstract or highly non-linear objective functions (e.g., complex risk metrics in portfolio optimization), does the "simplification" still result in a query that a general-purpose LLM can reliably reason about?

**Limitations:**

yes

**Strengths And Weaknesses:**

Strengths: the simplification-based verification paradigm is a new idea for LLM modeling and allows practitioners to automate optimization modeling with much higher confidence. The ability to detect subtle misformulations in constraints or objectives is a major step forward for critical applications in logistics, energy, finance, etc.
Weaknesses: the method is significantly more expensive in terms of Tokens and latency compared to tranditional direct prompting or basic self-debugging methods. This may limit its use to high-value or offline optimization tasks. And several bugs are remained to be solved as noted by the authors, e.g. if the LLM completely ignores a part of the problem description, the verification system might still pass the code as "internally consistent.

---

> ### Author Rebuttal · Authors · 2026-03-31
>
> Many thanks to the reviewer for the very helpful and constructive feedback.
>
> **Q1/Weaknesses**
>
> **Optimization opportunities**. While our focus in this work was on accuracy and precision/coverage of verification, we agree that optimizations to address the higher compute costs is an important direction, and that the *modular*  nature of our verification approach  presents  many such opportunities as the reviewer notes. In particular, we agree that prioritizing “risky” components is a promising direction, and in fact, one concrete way to estimate risk is via compositional consistency across sampled programs: constraints that occur consistently across all sample programs or decision variables that yield the same values across all samples may be treated as lower risk as opposed to constraints or variables that show more variation across samples, and  verification can be prioritized accordingly,
>
> Another adaptive optimization that we have already implemented is early stopping in the program sampling itself: rather than always sampling the full (K =10) programs, we stop early if  verification completely succeeds on a sample program. In a subset experiment (100 NL4Opt cases with GPT-4o), this yields an average number of LLM calls of  37.8, with ~40% of cases requiring fewer than 20 calls, showing that the framework already adapts computation based on problem difficulty, using less computation for easier cases and more for ambiguous ones. We also agree that the simplified nature of verification queries makes them amenable to smaller/cheaper LLMs to provide further potential cost reductions. All of these adaptive optimization strategies are very interesting future directions.
>
>
> **Cost-controlled comparison**. We agree that cost-controlled comparison with test-time scaling methods is valuable, and have therefore conducted an additional experiment comparing VeriSimpl with two common scaling strategies under matched (and even higher) compute budgets: (i) self-consistency (SC) with majority voting (sampling multiple programs and taking the most frequently occurring answer), and (ii) SelfDebug with extended repair attempts. On our subset of 100 LPWP instances using GPT-4o, VeriSimpl used an average of 37.8 LLM calls per problem (min 13, max 89). Hence for a conservative comparison, we allowed significantly higher budgets for the baselines on the same subset (100 LLM calls per problem, which exceeds the maximum number of LLM calls used by VeriSimpl in any case). We ran SC with 100 sample program generations (SC_100) and SelfDebug with up to 100 repair attempts (SelfDebug_100). The resulting accuracies:
>
> - BaseLLM: 75%
> - SelfDebug_100: 76%
> - SC_100: 77%
> - VeriSimpl: 89%
>
> Hence despite using substantially fewer LLM calls on average, VeriSimpl significantly outperforms both methods, indicating that the gains come from our verification-based technique rather than just more compute. We also note that unlike methods such as SC which always use the fixed compute budget on all questions, our method is adaptive in that it utilizes much less compute for simpler cases (40% of cases invoked less than 20 LLM calls).  Moreover, apart from pure accuracy improvement, our method also provides a high-precision verification signal and our verification queries  also provide finer-grained diagnostic feedback  (specific test failures can identify problematic constraints or objective misformulations)  which standard scaling approaches do not offer.
>
> Please also see our response to reviewer **qLjZ  (W1)** on addressing missing constraints from NL description.
>
> **Q2**
>
> While we have not yet evaluated highly non-linear settings such as portfolio optimization, we agree this would be an interesting direction to explore. Since the core principle of our approach is to reduce global reasoning over high-dimensional interactions into localized reasoning tasks, this can apply broadly to any optimization problems where there are multiple interacting constraints and decision variables, which is typical in practice. For example, even with a complex or non-linear objective, if fixed values for all decision variables are provided then that reduces the task to only computing the objective value from concrete inputs, which is significantly more tractable than solving the full problem from scratch (our method does not require the LLM to reason about the full objective symbolically with multiple unknowns). That said, the ultimate effectiveness of this reduction still depends on the residual complexity of the simplified query - for highly abstract or complex objectives the remaining reasoning may still be challenging for current LLMs, and this merits further investigation. Thus while our approach systematically reduces complexity along key dimensions, its effectiveness is also coupled with the capabilities of the underlying model. As LLM reasoning improves, we can also expect the benefits of the approach to extend naturally to more complex domains.

---

> > ### Author Rebuttal · Reviewer_twEV · 2026-04-03
> >
> > Thanks for the authors' response, after reading the rebuttal, I think it is appropriate to maintain my current score.

---

### Decision · Program_Chairs · 2026-04-30

**Decision:**

Accept (regular)

**Comment:**

This paper attempts to make llm generation of optimization more robust by automatically creating simple diagnostics tests to verify the llm output. The empirical results confirm the efficacy of the proposal, though the reviewers commented that the paper is hard to follow so I encourage the authors to clean up the presentation, and especially detailed the experimental set-up, for the camera-ready.